# Spectroscopy Approach for Highly-Efficient Screening of Lectin-Ligand Interactions in Application for Mannose Receptor and Molecular Containers for Antibacterial Drugs

**DOI:** 10.3390/ph15050625

**Published:** 2022-05-19

**Authors:** Igor D. Zlotnikov, Elena V. Kudryashova

**Affiliations:** Department of Chemistry, M.V. Lomonosov Moscow State University, Leninskie Gory, 1/11B, 119991 Moscow, Russia; zlotnikovid@my.msu.ru

**Keywords:** targeted delivery, levofloxacin, eugenol, double inclusion complex, cyclodextrin, molecular container, synergism, allylpolyalkoxybenzenes

## Abstract

Rational search of a ligand for a specific receptor is a cornerstone of a typical drug discovery process. However, to make it more “rational” one would appreciate having detailed information on the functional groups involved in ligand-receptor interaction. Typically, the 3D structure of a ligand-receptor complex can be built on the basis of time-consuming X-ray crystallography data. Here, a combination of FTIR and fluorescence methods, together with appropriate processing, yields valuable information about the functional groups of both the ligand and receptor involved in the interaction, with the simplicity of conventional spectrophotometry. We have synthesized the “molecular containers” based on cyclodextrins, polyethyleneimines (PEI) or spermine with mannose-rich side-chains of different molecular architecture (reticulated, star-shaped and branched) with variable parameters to facilitate delivery to alveolar macrophages. We have shown that synthetic mannose-rich conjugates are highly affine to the model mannose receptor ConA: *K*_d_ ≈ 10^−5^–10^−7^ M vs. natural ligand trimannoside (10^−5^ M). Further, it was shown that molecular containers effectively load levofloxacin (dissociation constants are 5·10^−4^–5·10^−6^ M) and the eugenol adjuvant (up to 15–80 drug molecules for each conjugate molecule) by including them in the cyclodextrins cavities, as well as by interacting with polymer chains. Promising formulations of levofloxacin and its enhancer (eugenol) in star-shaped and polymer conjugates of high capacity were obtained. UV spectroscopy demonstrated a doubling of the release time of levofloxacin into the external solution from the complexes with conjugates, and the effective action time (time of 80% release) was increased from 0.5 to 20–70 h. The synergy effect of antibacterial activity of levofloxacin and its adjuvants eugenol and apiol on *Escherichia coli* was demonstrated: the minimum effective concentration of the antibiotic was approximately halved.

## 1. Introduction

Ligand-receptor interactions play a significant role in the functioning of many biological systems [1], especially the immune system [2]. Regulation of the complex formation processes of proteins with bioactive agents can be considered as a potential approach to targeted impact on tissues and cells. There is widespread application in medical biochemistry of linking high-affinity ligands to receptors, for example, galactosylated drug carriers to the asialoglycoprotein receptor [3], or drug carriers functionalized by monoclonal antibodies to the E-selectin [4,5], which is expressed by endothelial cells during an acute or chronic inflammatory reaction. This is due to the high therapeutic potential of these systems in the treatment of a number of diseases, including oncological, neurodegenerative, autoimmune diseases [6].

Macrophages as key cells of the immune system are a potential target for the delivery of antibacterial drugs or agents capable of regulating the activation status of macrophages, depending on the treatment objectives [7,8]. Macrophages express a large number of receptors [9], including Toll-like and scavenger receptors, integrins and the most significant in our research field, C-type lectin domains (CTLDs). Clustering of 8 CTLDs provides specific recognition of pathogens by mannose receptors, including CD206 [10]. CD206 is a transmembrane protein (175 kDa) that recognizes Man, Fuc and GlcNAc residues of oligosaccharides of the cell wall of pathogenic microorganisms [11]. Carbohydrate binding occurs due to the C-type region consisting of 8 domains; however, the greatest contribution to the ability to bind carbohydrates in isolation from other parts is shown only for the fourth carbohydrate–recognizing domain (CRD4) [12,13,14]. Therefore, determining the parameters of a carbohydrate ligand for high-affinity interaction with CD206 or its domains is an urgent biochemical issue. Various methods of studying receptor-ligand interactions on the example of the mannose receptor are presented in the literature: X-ray crystallography of trimmed CRD4 [13], oligosaccharide binding competition assays [13,14,15,16], NMR shifts [15,16]. However, these methods are not express and require strict validation of the experimental methodology. In addition, due to the limited availability of the CD206 receptor [13], the range of applicable methods is limited; also high throughput screening analysis of ligands is not available. Accordingly, in such cases, a model lectin ConA can be used, which is more suitable for screening purposes. The relevance of ConA as a CD206 model has been widely discussed in the literature and studied by us through computer modeling of 15 ligands [17]. We have previously shown the similarity of the structural organization of the binding sites in ConA and in CD206 (CRD4) with high correlation of the values of the free energies of complexation (r > 0.9).

ConA is an easily accessible lectin having a carbohydrate-binding domain structure similar to CD206 and a general similarity in carbohydrate-binding ability [8,17,18,19,20,21]. According to the literature, the CD206 domain shows a high similarity with mannan binding lectin A, which, in turn, is similar to ConA in terms of carbohydrate binding [22]. This is confirmed by numerous in vivo and in vitro studies [20]. Mannose-containing ligands, pre-selected by ConA assay, show high affinity (by flow cytometry) to CD206+ cells [23,24]. Similarly, mannose-containing ligands, pre-selected by ConA assay, are absorbed by macrophages via a receptor-mediated internalization mechanism [7], and are bio-distributed to lungs due to adsorption by macrophages [25]. All this data supports the relevance of using ConA as a mannose receptor model. 

A wide range of methods can be applied to the study of ConA–ligand interactions [8]: isothermal titration calorimetry [21,26], fluorescence methods [18,27], FTIR spectroscopy [28], Landsteiner inhibition method [29]. In silico methods are no less valuable and highly informative [30,31].

However, the known techniques have good reliability for mono- and oligosaccharides. Moreover, in the case of polymer molecules or their conjugates, the study of receptor-ligand interactions is quite a difficult task, due to the complex 3D architecture and interpretation of data, for example, in isothermal calorimetry effect of dissolution of the polymer system itself. It is necessary to reproducibly and reliably determine the parameters of conjugate (ligand) interaction with receptors with minimal consumption of proteins and ligands. A wide range of methods are described in the literature, but not all can be applied to the systems under consideration.

The proposed spectral approaches (IR, fluorescence) make it possible to increase sensitivity to the concentrations of the receptor and ligands, which means to reduce the consumption of expensive reagents, and, more importantly, to obtain valuable information about the mechanism of complexation (microenvironment of amino acid residues, protonation of carboxyl groups, formation of hydrogen bonds and electrostatic interactions).

The methodology for investigating the effectiveness of lectin-ligand interactions has been optimized here in the application of newly developed highly-mannosylated multifunctional conjugates for targeting macrophages based on cyclodextrin with polyethylenimine or spermine and mannose. Moreover, we have been studied the drug loading and release depending on the nature of the ligand, molecular weight, degree of branching, molecular architecture—using fluorescence quenching and polarization, UV and Fourier-transform infrared spectroscopy (FTIR).

Cyclodextrins are introduced into the conjugate so that the system is able to form inclusion complexes with biologically active substances and drugs [32,33,34,35]. At the same time, cyclodextrins protect them from oxidation, hydrolysis, enzymatic destruction, excessive hygroscopicity [36], and, in addition, cause increased penetration of drugs through biological membranes, which is extremely important in the field of drug delivery [37]. To achieve an additional effect, cyclodextrins were modified with spermine molecules or attached to the polymer chain of PEI, which is often used for targeted delivery [38,39]. This brings the positive charge of the conjugate, and the ability to hold the therapeutic agent more firmly and release it in a prolonged manner due to the polymer mesh [40]. In addition, the presence of amino groups allows modification of mannose for high-affinity interaction of ligands with mannose receptors.

Currently, a promising direction in the treatment of lung infections is research related to the use of combined antibacterial drugs. Thus, the effectiveness of chemotherapy in people with severe lung infections (primarily tuberculosis) is due to the use of new generation drugs or adjuvants (enhancers) with a fundamentally different mechanism of action, for example Bedaquiline (Sirturo, Janssen Biotech, Inc., Philadelphia, PA, USA) in combination with the main antibacterial agent (such as levofloxacin) [41]. It is assumed that various adjuvants can increase the binding strength of levofloxacin to bacterial DNA gyrase, as showed for ginsenoside [42].

We chose the model fluoroquinolone [37,40,43,44,45], levofloxacin and the promising adjuvant eugenol (clove oil) [46,47,48], as expected, which showed the effect of synergism, which would reduce the dosage of the main “poisonous” drug [49,50,51]. For these “cargos” to function, it is necessary to dissolve them by loading cyclodextrin into the tori and deliver them to the alveolar macrophages of the lungs (for example, intranasal or inhalation forms).

Thus, in this work: (1) we synthesized promising drug delivery systems with high affinity to mannose receptor macrophages using alternative methods for obtaining mannose clusters instead of very expensive oligomannosides, (2) determined the parameters for the inclusion of levofloxacin and eugenol drugs, (3) presented a combined formulation of these drugs, and (4) developed effective spectral methods for studying receptor-ligand interactions.

## 2. Results and Discussion

### 2.1. HPCD-PEI-Man & HPCD-Spermine-Man Conjugates Synthesis and Characterization

The main goal of the work is to develop an optimal molecular container for targeted delivery of levofloxacin to mannose receptors of macrophages and to work out the spectral approach for high-throughput screening of lectin-ligand interactions using concanavalin A as a model lectin. Using different ratios of components, varying the conditions and the reaction sequence, we obtained 9 mannosylated HPCD conjugates with different molecular architecture (Table 1), with spermine as spacer for mannose clusters (high affinity for CD206) or with PEI of various molecular weights and degrees of branching and cyclodextrins’ tori for drug loading—according to the schemes shown in Figure 1 and Appendix A.

Such variability of the obtained products is very promising in the field of synthesis of targeted drug delivery systems with high affinity for mannose receptors. Moreover, it is possible to adjust the capacity for loading the drug due to the amount of HPCD or polymer chains, vary the physico-chemical parameters (Table 1: particle size, hydrophilicity, charges, zeta potential, degree of mannosylation) to optimize the platform for a specific task.

All synthesis started with the activation of secondary hydroxyl groups of cyclodextrin with CDI followed by cyclodextrins’ modification with the primary and secondary amines of spermine or PEI (Figure 1a,b and Appendix A) based on the reactions described in works [52,53,54,55,56] with author’s modifications. Finally, reductive mannosylation of amino groups has been performed. The successful formation of conjugates was demonstrated by Fourier transform infrared (FTIR) spectroscopy, as shown in Figure 2 and Appendix A. Let us consider the characterization of the structure by the example of a conjugate HPCD-spermine-Man-**7**. In the HPCD spectrum (Figure 2a), characteristic strong absorption bands of valence vibrations of C–O and C–O–C bonds 1030–1150 cm^−1^ are observed. After modification by spermine (Figure 2b), hydroxypropyl groups become cross-linked with carbonyl amide bond –O–C(=O)–NH–. Thus, corresponding peaks become narrower and more intense (Figure 2c—1046 and 1085 cm^−1^) due to relative intensification of the peak of C–O bonds in the glucopyranose residues of CD themselves and also due to the C-N-C bonds of spermine molecules. In the region of 2800–3000 cm^−1^, characteristic sharp peaks of symmetric (2855 cm^−1^) and antisymmetric (2926 cm^−1^) valence vibrations of CH_2_ groups in spermine (Figure 2b) are superimposed on a wide absorption band of valence vibrations of charged amines NH^+^, NH_2_^+^, NH_3_^+^ from spermine. The most important intense peak in the IR spectrum at 1630–1650 cm^−1^ corresponds to C=O oscillations in urethane groups –O–C(=O)–NH– formed as a result of crosslinking of OH groups of CD and NH groups of spermine with carbonyldiimidazole. The spectrum of the mannosylated conjugate (Figure 2d) is characterized by reducing the absorption of vibrations of amino groups in the region of 2700–3000 cm^−1^, due to their modification and “fixing” the mobile hydrogen atoms; by broadening and changing the absorption form of C-O-C and C–O valence vibrations due to the introduction of mannose into the system; and by overlapping the peaks of C-N-C and C-O-C bonds. Determination of the ratio mannose:cyclodextrin was carried out by decomposing a section of the spectrum 980–1130 cm^−1^ into components, using FTIR spectra of the constituents (Appendix A). The degree of mannosylation in conjugates determined by FTIR spectroscopy is in good correlation with the data obtained by visible spectroscopy (by titration of amino groups in conjugates with TNBS) as well as by fluorimetric titration of amino groups with OPA (Appendix A).

The proposed conjugate’s structures and molecular weights are identified on the basis of conditions and sequence of synthesis reactions, the ratio of components, IR spectra, data on the number of mannosylated amino groups (Table 1). Conjugates **1–3** (Appendix A) are oligomeric cyclodextrin’s meshes with mannose clusters; they have low-negative zeta-potential (–3–4 mV, because charged groups are virtually absent) and a high hydrodynamic diameter from 330 to 780 nm in accordance with the increase in molecular weight from 6 to 9 kDa and most significantly—the number of mannose clusters (mannose molecules from one spermine molecule) on spermine spacers from 3–5 to 10–15. This size (780 _HM_) includes a shell of water around the conjugate, so for “sugar” conjugates, the size turns out to be large (as earlier reported for mannan 530–720 nm [57]). Products **1–2** contain linear and pyranose mannose residues that have been attached to activated OH groups of cyclodextrin. Product **3** is a highly mannosylated HPCD mesh, which was synthesized using a significant excess of CDI (40 eq.), which prevented crosslinking of HPCD tori. Conjugates **4–6** (Table 1 and Appendix A) of PEI35-series are polymer chains grafted with cyclodextrins, compacted with medium hydrodynamic diameter of about 55 nm. Ligands **4** and **6** have a positive zeta-potential due to the amino group of PEI35, but in conjugate **5** there are practically no free amino groups; moreover, a negative charge is possible due to the presence of COO^–^ groups on CD remaining after activation with carbonyldiimidazole, which has less effect on other conjugates with a smaller number of CD. In addition, we obtained star-shaped and probably highly hydrated (150 and 67 nm) conjugates **7–8** and huge branched polymer **9** (Table 1 and Figure 1 and Appendix A): in the center there is one or more HPCD’s tori with spermine (4.5 eq) attached to them, or branched PEI 1.8 and 10 kDa–6.1 eq and 0.5 eq. respectively. For conjugates **7–9,** zeta-potential close to 0 corresponds to an almost complete modification of the amino groups of PEI or spermine and effective crosslinking (there are no non-reacted COO^–^).

The obtained data on the structure of the synthesized conjugates were confirmed by NMR spectroscopy. In the 1H NMR spectra (Figure 2e–h) of conjugates, both peaks of the initial components and new signals related to urethane bonds −N(H)−C(O) (formed as a result of the interaction of activated HPCD with spermine or PEI) were detected [58,59]. The signals in the range of 4.0–3.2 ppm and the singlet of 5.09 ppm correspond to H2–H5 and H1 of the D-glucopyranose or D-mannose link of the oligosaccharide, respectively [24]. The presence of spermine in conjugate **7** is confirmed by a 1.0–1.1 ppm doublet and 1.78 and 2.11 pm singlets in the 1H NMR spectrum (Figure 2g). The presence of PEI and the formation of urethane bonds—HPCD-CH2-O-C(O)-NH-CH2- in conjugates is confirmed by the signals in the range of 2.8–3.0 pm [40,54]. The number of urethane bonds in terms of one cyclodextrin torus is largest for conjugates **5**, **6** and **8**, and least for conjugate **9**, which is consistent with the data on the ratio of components determined by IR spectroscopy.

### 2.2. HPCD-PEI-Man & HPCD-Spermine-Man Conjugates Binding with a Model Mannose Receptor ConA

The requirements imposed on the drug carrier include high affinity for the mannose receptor of macrophages. Relevant data on the affinity for mannose receptors of CD206 macrophages can be obtained using model experiments with lectin concanavalin A, because of a high correlation of carbohydrate binding capacity with the fourth domain (CRD4) of CD206 as was shown by in silico methods [17]. Therefore, it is necessary to have a robust, reproducible and accessible method for studying protein-receptor binding [8,18], which is generally complicated for structurally sophisticated ligands. A small number of techniques for similar systems are described in the literature: isothermal calorimetry of titration on complexation of mannosylated multivalent carbohydrates [26] or cyclodextrins with mannose clusters [60] with ConA, with dependence of hemagglutination inhibition on neoglycopolymer length [61]. In this paper, a promising approach has been suggested to study ligand–receptor binding using Fourier infrared spectroscopy (FTIR) and Trp or FITC fluorescence quenching and anisotropy.

#### 2.2.1. Fluorescence Quenching and Anisotropy

We proposed a technique allowing to reliably determine the efficiency of binding without the introduction of fluorescent labels into the ligand (as described in the literature [23,53]). The technique is based on the fluorescence quenching of tryptophan residues in a protein when binding to a ligand [18], as well as an increase in the anisotropy of Trp fluorescence due to an increase in the molar mass of the ConA–ligand complex in comparison with an unbound receptor. The requirement for precise and correct determination of dissociation constants is a ligand mass (for fluorescence anisotropy) of at least 10 kDa (about 20% of dimeric ConA 53 kDa). An alternative approach that increases sensitivity and removes the restriction on molecular weight is possible—the introduction of a fluorescent label (for example FITC) into a conjugate. Data of a model experiment with labeled cyclodextrin are shown—Appendix A. However, for the systems under consideration, FITC (and other fluorescent molecules) can form an inclusion complex with neighboring CD tori, which can affect the K_d_ values. This fact complicates the interpretation of the data significantly. Thus, the use of the protein’s own fluorescence simplifies the screening of ligand-receptor affinities.

The interaction of “heavy” conjugates (PEI group with molecular weight in the range from 20 to 115 kDa) with ConA was studied by changing the fluorescence anisotropy (r), using the signal of the aromatic tryptophan residue in ConA. The adsorption curves (Figure 3a) are typical saturation lines, most accurately approximated by the Hill (3) and Boltzmann (4) models. The magnitude of the change in r correlates with the molar mass of the ligand (from 50 to 100% of initial anisotropy value of protein). The processing of sorption curves in Hill coordinates (10) leads to an S-shaped dependence (Figure 3b) [62], which can be approximated by a single straight line (short points for the blue curve) to determine the apparent dissociation constant of the ConA-ligand K_obs_ complex or to isolate three linear sections [63] and consider them separately to calculate K_d_ in several binding modes, to identify positive/negative cooperativeness (dash lines (– –) for red curve in Figure 3b). Thus, for the HPCD-PEI35-Man-**5** ligand, K_obs_ = (60 ± 9) μM, while the maximum affinity for the most “successful spatial configurations” of ligand molecules (for the same ligand) (in the presence of positive cooperativeness [8,18]) increases: K_max_ = (5.0 ± 0.3) μM.

To prove that mannosylated conjugates interact specifically with the model ConA receptor precisely at the mannose binding sites, a competitive complexation of HPCD-PEI35-Man-**5** was carried out in the presence of 0.5 M mannose (Figure 3a). Titration of the receptor solution with a ligand is initially characterized by a decrease in the anisotropy of tryptophan fluorescence due to changes in the microenvironment from a viscous mannose to a conjugate. 0.5 M mannose solution increases the viscosity (viscosity is about 1–1.5 mPa·s) and thereby the anisotropy of Trp in ConA from 0.08 to 0.10. Moreover, the presence of concentrated mannose changes the form of the binding curve to a gentler one, which indicates a competitive mechanism of complexation. However, ligand binding is noticeable only at a threefold molar excess: C_lig_ = 44 µM, C_Man_ = 460 mM, which indicates a high affinity binding of the mannosylated polymer, 3–4 orders of magnitude superior in affinity to mannose (K_d_ 0.1 µM versus 1 mM, respectively [8]).

The interaction of “light” conjugates (spermine group with molecular weight in range from 6 to 9 kDa) with ConA was studied by changing the fluorescence intensity F of the aromatic tryptophan residues (the fluorescence of tyrosine residues under these conditions is significantly less: λ_max_(Tyr) = 303 nm, λ_max_ (Trp) = 340 nm, λ_max_ (observed) = 335 nm) in ConA (Appendix A for mannan). The analysis was carried out similarly to the recently described methods for mannosylated chitosans [18]. Despite the fact that quenching the tryptophan fluorescence in a protein is a fast affinity screening method, there is a disadvantage: at high concentrations of the ligand, dynamic factors can play a significant role. Its accounting can be carried out according to the modified Stern–Vollmer equation [18] by highlighting the static and dynamic (K_st_ and K_din_) components: F_0_/F = 1 + (K_din_ + K_st_)[Q] + K_din_ ∙ K_st_ ∙ [Q]^2^, where F_0_ and F are the initial and observed fluorescence intensities of the ConA-ligand complex, respectively, and [Q] is the equilibrium concentration of the fluorescence quencher (conjugate). According to this equation, it was possible to determine the dynamic contribution of tryptophan fluorescence quenching in ConA by mannose molecules (0.5 M) in the case of competitive binding of the HPCD-PEI35-Man-5 ligand (Appendix A): K_st_ = 3060, K_din_ = 0.06—superiority in affinity of the polymer ligand by 4–5 orders of magnitude.

So, the methods of polarization and fluorescence quenching are applicable to the studied systems to determine the affinity of the ligand to the receptor, and moreover, as will be shown later, to study the interaction of molecular containers with drugs.

#### 2.2.2. Fourier-Transform IR Spectroscopy (FTIR)

FTIR is highly informative in relation to the studied ligand-receptor interaction, because it gives information about the spatial arrangement polymeric chains, the changes in the protein tertiary and secondary structure, the microenvironment of certain functional groups or the formation of non-covalent interactions (van der Waals, electrostatic, hydrophobic) by changing the following characteristic regions (both in proteins and polymers): amide A (valence vibrations of N–H bond atoms), amide B, amide I (contribution of valence vibrations of C=O bonds—80%; and C–N bonds—about 15%), amide II (main contribution of deformation vibrations of N–H bonds (80%) and minor contribution of valence vibrations of C–N bonds—about 15%) and amide III (the fallback option is difficult to interpret and overlaps with other peaks).

Recently we studied in silico the detailed mechanism of these interactions [17] (Appendix A), using the interaction of trimannoside with ConA as the example. The “amide” atoms of the binding site are involved in complexation-Asn14 (N)-Man (O3), Asp16 (H)-Man (O5), Arg222 (H)-Man (O2). Thus, changes in the amide II region are expected when the ligand binds to the protein. Moreover, carbonyl oxygen atoms of the Gly-peptide bond are also involved in complexation [17]. Experimental data on changes in the Amide I and Amide II regions (ConA spectrum) upon binding to the ligand HPCD-spermine-Man-**1** (Figure 4a) confirm the formation of point interactions of oxygen atoms of the carbonyl group and hydrogen or nitrogen atoms in the amide with ligand donors/acceptors. The microenvironment of the peptide bond atoms changes (predominantly in the binding site of the ligand), which is reflected in the IR spectrum: the intensity of the peaks of amides I and II decreases (Figure 4a). Computer modeling [17,64] and crystallographic methods [14,65,66] allow us to determine the point interactions of ligand and receptor atoms, the formation of which leads to a change in the energies of certain amino acid residues, which theoretically should be displayed in the IR spectrum. However, the analyzed peaks correspond to a whole set of bonds, and therefore, the target minor change (against the background of a huge number of other peptide bonds) is practically unnoticeable. Previously, it was proposed to normalize the peak of amide II to Amide I and only monitor the intensity [28], but this is not always correct. Therefore, we differentiated the spectra and obtained a more detailed picture in which energy-changing components can be distinguished. The dissociation constants were calculated based on the dependence of the peak position at 155(6–9) or 153(4–6) cm^−1^ in the second-order differential IR spectrum (Figure 4b). Differentiation allows you to select the target component of the analysis (N–H and C=O valence oscillations—1500–1575 and 1620–1650 cm^−1^, respectively)—in a narrow energy region (1–5 cm^−1^) and corresponding peptides.

Thus, we were able to identify shifts in the differential IR spectrum of the second order in the Amide II region at 1558 and 1535, 1532 cm^−1^. Changes are also observed in other peaks, but the most relevant data are believed to be obtained from the peak at the maximum position, the change in which corresponds to the energy gain during complexation with ligand. Analysis of the 2nd derivative of IR spectra leads to hyperbolic curves (Appendix A), showing the dependence of the analytical signal on the concentration of the added ligand to the receptor. Linearization is most representative in Hill’s coordinates (Appendix A), which, in addition to the dissociation constant, allows us to determine the effect of cooperativeness during ligand binding.

Thus, the approaches discussed above, based on the use of IR spectroscopy and fluorescence analysis methods, make it possible to study the receptor-ligand interaction even for complex conjugates of a wide range of molecular composition, which is difficult when using classical methods such as isothermal calorimetry [21,65]. The following is a discussion of the results of experiments to determine the affinity of ligands with ConA.

#### 2.2.3. The Dependences of Conjugates’ Affinity to the Model ConA Receptor on the Parameters of the Molecular Containers

One of the tasks of the work is the selection of optimal parameters (Table 1 and Table 2) for drug delivery systems to macrophages by varying the composition, molecular architecture and molar ratio of components of ligands. Therefore, we determined the dissociation constants of conjugate complexes with the ConA mannose receptor (Table 2). The “light” star-shaped conjugates HPCD-spermine-Man-(**3,7**) were the most affine in comparison with systems with polymer substituents PEI 1.8, 10 and 35 kDa. With an increase in the molecular weight of the polymer from 1.8 to 35 kDa, the affinity to the mannose receptor increases, provided the optimal configuration of the components (in a series of ligands **6** > **9** > **8**). It is noteworthy that conjugates with PEI are characterized by several binding modes due to the possibility of ligand binding by a low-affinity part (cyclodextrin) and a high-affinity mannose cluster on a spermine or a branch of PEI—*K*_max_ value. During the experiment, an average pattern is recorded—*K*_obs_. Positive cooperativeness is observed for all systems, since in Hill coordinates the slope of the straight line is greater than 1. The degree of cooperativeness in Equation (3) increases with increasing conjugate mass and affinity to the receptor. In the case of a branched conjugate with a PEI 10 kDa (* in Table 2) *n* > 3 is observed, which indicates multivalent ligand binding at several sites of ConA (one dimeric protein or several globules), providing nanomolar total dissociation constants of complexes—supramolecular ensembles of protein and ligand. Let us discuss the influence of molecular architecture on affinity to mannose receptors. The HPCD-spermine-Man-**1** and **2** ligands (Appendix A) are crosslinked meshes of cyclodextrins with mannose residues, *K*_d_ of the order of 5·10^−5^ M; and ligand **3** (Appendix A) of a similar composition, but with a lower degree of crosslinking of CD and mannose clusters on spermines is half an order superior to the high-affinity mannan: 5·10^−7^ M vs. 1 µM. Moreover, ligand **7** (Figure 1a) with star-shaped architecture, even more affine to mannose receptor: *K*_d_ ≈ 2.5·10^−7^ M. In the series of HPCD-PEI35-Man conjugates (**4–6**): the composition of ligand number **6** (1:2 CD/PEI) is the most optimal from the point of view of binding to the receptor; however, ligand **5** (8:1 CD/PEI) has a large number of CD tours, which is likely to increase loading capacity for the drug, unlike samples **4** and **6**, although it remains affine to mannose receptors: *K*_d_ of the order of 5·10^−5^ M and up to 5·10^−6^ M—affinity in the optimal mode. Conjugates **8** and **9** with branched PEI are highly affine to ConA and are characterized by a higher relative mass content of CD due to a decrease in the mass of the polymer. Thus, we have obtained conjugates comparable and even superior in affinity to the natural trimannoside ligand to the mannose receptor ConA (*K*_d_ ≈ 10 μM [8,21,28]) and, moreover, superior to the high affinity polymer mannan (*K*_d_ ≈ 1 μM [66]). Thus, the condition of high specificity of drug delivery systems to mannose receptors of macrophages is fulfilled. The physicochemical aspects of the interaction of conjugates with levofloxacin and eugenol, a promising combination of therapeutic agents, are described below.

### 2.3. Levofloxacin and the Adjuvant Eugenol Interactions with HPCD-PEI-Man and HPCD-Spermine-Man Conjugates

Next, experiments are shown to study the affinity of the interaction of conjugates with fluoroquinolones and eugenol by fluorescent methods, FTIR and UV spectroscopy.

#### 2.3.1. Levofloxacin Fluorescence Quenching and Polarization

Fluorescent methods are well suited to studying both protein-ligand interactions and the binding of aromatic molecules to molecular containers. In our case: levofloxacin interacts with mesh HPCD-spermine-Man-**2** and polymer HPCD-PEI35-Man-**4** conjugates (Appendix A). However, the sensitivity and magnitude of the change in the analytical signal increases in comparison tryptophan fluorescence: anisotropy of levofloxacin changed from about 0.02 to 0.15 units; fluorescence quenched from 650 to 250 units. If, in the case of protein, the most convenient and accurate method was polarization analysis, then the interaction of aromatic molecules with conjugates based on cyclodextrins and polyethyleneimines were described more accurately as the quenching of fluorescence.

The mesh conjugate HPCD-spermine-Man-**2** (Appendix A) interacts with 20 levofloxacin molecules maximally (determined by fitting model—see Methods), holding them into the inner cavity of the CD (tori) due to hydrophobic interactions, as well as the interactions with the polymer matrix by non-covalent interactions (hydrogen bonds, electrostatic) contributing to the stability of the complex with levofloxacin.

In the literature, loading of levofloxacin in poly(lactic-co-glycolic acid) and chitosan nanoparticles [67,68], star poly(e-caprolactone) scaffolds [69] was described. The studied conjugates are also able to effectively incorporate drugs. The dissociation constants of Lev –HPCD-spermine-Man-**2** are about 320 μM (three times better than a simple HPCD [70]) with the simultaneous increased capacity of the drug. The cyclodextrin conjugate with a heavy PEI of 35 kDa (Appendix A) retains up to 40–50 levofloxacin molecules (*K*_d_ ≈ 21 μM—more affine than spermine’s conjugate) presumably to a greater extent due to their interaction with polymer chains. Note that the quenching of fluorescence, as expected, is expressed by a Hill type dependence (Equation (3)) while, in contrast, the anisotropy increases almost linearly or exponentially (Appendix A)—due to the formation of molecular aggregates of the ligand’s molecules.

#### 2.3.2. Eugenol and Levofloxacin: Fourier-Transform IR Spectroscopy Approach

The fluorescent approach to study the interaction of fluoroquinolones with conjugates allows us to determine the dissociation constants and the loading of the drug, but does not provide information about the molecular mechanism of the interaction of an aromatic molecule with the delivery system. As a complementary method for determining the molecular details of the formation of drug inclusion complexes with the studied polymer ligands, the FTIR spectroscopy is proposed, allowing monitoring of the involvement of specific functional groups in ligand interactions on the part of the drug and drug binding sites on the part of ligands. Moreover, IR spectroscopy, unlike most other methods, makes it possible to study the interaction of multicomponent systems due to the large number of analytically significant peaks, which provides information about the state of each component.

##### Levofloxacin

The parameters of complexation of levofloxacin and the eugenol adjuvant with molecular containers were determined by FTIR in solutions (Figure 5). The most noticeable changes were in the microenvironment of the quinolone structure of levofloxacin (corresponding peaks of aromatic C-C bond fluctuations at 1474 and C-N bond 1398 cm^−1^). As previously reported [44], Lev (Figure 5a, insert) exists in zwitterionic form in water solutions and consequently the molecular structure of Lev–CD complex with amino and carboxylate groups of Lev outside the CD cavity is stabilized by forming of intermolecular hydrogen bonds with amino group of spermine and hydroxyl groups of cyclodextrin. The appearance of shoulder peaks in the range of 1425–1465 cm^−1^ confirms several variants of the location of levofloxacin in relation to the cyclodextrin cavity and also corresponds to the Lev–spermine interaction (control experimental spectra—Appendix A).

During complexation with the HPCD-spermine-Man-**1** molecular container, the intensity of the 1474 cm^−1^ peak decreases by about 40%, and shoulder peaks also appear (Figure 5), which indicates the inclusion of Lev in the oligomeric mesh or CD cavities. The shift of this peak to the region of low wave numbers corresponds to the interaction of Lev with spermine (Appendix A). An increase in intensity of the peak of aromatic C-N bonds at 1398 cm^−1^ is due to the effects of the polymer chains^,^ because formation of “guest-host” complexes Lev with monomeric CD, in contrast, resulted in decreasing of intensity (Appendix A) [70,71]. 

The dissociation constants of the complexes of molecular containers with levofloxacin were calculated based on changes in the intensity of peaks at 1474 cm^−1^ (Figure 5a, Appendix A, Table 3), since the most pronounced changes are observed in this area. Constants of the order of 0.5 mM are characteristic for conjugates of the «spermine» series **1–3**: at the same time, mesh ligands **1–2** include Lev more efficiently than a highly mannosylated branched ligand **3**. However, all these conjugates are capable of “loading” more than 10–15 Lev per polymer molecule. Polymer systems **5** and **6** have significantly higher parameters of the loading capacity. In this case, conjugate **5** interacts with levofloxacin 2 orders of magnitude more affine due to the optimal PEI/cyclodextrin ratio of the order of 1:10. The linear polymer can fold into a molecular tangle: therefore, the drug is fixed more firmly in the cavities of the CD: *K*_d_ ≈ 5 μM.

Levofloxacin is included in conjugates with high entrapment efficiency (EE) (up to 98% of the initial amount Table 3), while the load capacity by weight of molecular containers for the drug is also high: mostly in the range from 52 to 86%. The smallest EE is found in the case of heavy polymer molecules. Conjugates **1**, **7** (oligomeric grid CD 6 kDa and star-shaped form, respectively) demonstrate the highest parameters of loading capacity and entrapment efficiency (EE and LC) for both levofloxacin and eugenol. Conjugate **7** combines the valuable properties of a drug container and a high affinity to mannose receptors due to presence of the trimannoside analogue on spermine spacer. 

##### Eugenol

Eugenol (EG) is an oily liquid (ρ = 1.065 g/mL), practically insoluble in water, which does not allow the use of this substance for the preparation and use in aqueous forms of therapy. However, the developed molecular containers form inclusion complexes with aromatic hydrophobic molecules due to the hydrophobic cavities of cyclodextrin, significantly increase the solubility of eugenol and allow the creation of combined enhanced forms of fluoroquinolones. The interaction of eugenol with conjugates based on CD and PEI35 was studied by IR spectroscopy (Figure 5b). When in complex formation with a ligand, eugenol interacts with polymer chains or is included in the CD cavity, which is confirmed by an increase in the intensity of the corresponding characteristic peaks: C=C valence stretching (1638 cm^−1^), C–C aromatic (1515 cm^−1^), O–H bending (1366 cm^−1^), alkyl aryl ether C–O stretching (1200–1275 cm^−1^)—and probably EG passes from an emulsion state to a dissolved form. The dissociation constants of complexes of molecular containers with eugenol were calculated based on changes in the intensity of peaks at 1515 cm^−1^ in the IR spectra (Figure 5b, Appendix A, Table 3). The interaction of the oil with the HPCD-PEI35-Man-**5** conjugate is preferable due to the increased number of CD tori per polymer molecule (approximately 9 vs. 1–2 in the case of ligands **4** and **6**): the difference in equilibrium constants is half an order. However, the star-shaped cyclodextrin conjugate **7** demonstrated even superior EE in comparison with the polymer grafted conjugate **5** and, moreover, an order of magnitude increased LC due to a decrease in the molecular weight of the system. The loading capacity of eugenol is less than in the case of levofloxacin due to lower affinity of eugenol to conjugates. However, the calculated parameters characterize molecular systems as effective drug delivery systems.

Since we have successfully synthesized inclusion complexes of levofloxacin and eugenol in the tori of CD conjugates, then the task is to obtain double complexes of inclusion of both drug molecules at once for possible use as a combined drug in one vial.

#### 2.3.3. Eugenol and Levofloxacin Double Inclusion Complexes in Molecular Containers FTIR Microscopy Approach

We synthesized complexes of inclusion of eugenol and Lev in the polymer conjugate HPCD-PEI35-Man-**5** and the star-shaped conjugate HPCD-spermine-Man-**7**. The characterization of the composition and distribution of the components was carried out by IR microscopy. The distribution of the components of the double complex was studied using IR microscopy, which allows measurements with a spatial resolution of up to 5 microns in order to analyze the inclusions and defects in polymer systems. IR spectra are recorded at several points (40) and peak intensities are integrated at specified intervals, which are displayed in a “heat” map corresponding to the quantity of components.

Thus, the HPCD-PEI35-Man-**5** conjugate uniformly includes aromatic Lev and EG molecules (pink and white colors—Appendix A). Figure 6 shows the distribution of CD (a), levofloxacin (b), eugenol (c) in a double drug complex with HPCD-spermine-Man-**7** conjugate according to the integral intensity of the characteristic peaks of the C–O–C bonds, C–C_arom_ and C=O, C–C_arom_ bonds (d), respectively. Cyclodextrin in the composition of the star–shaped conjugate is presented in large quantities and distributed evenly (pink and white colors in Figure 6a). Levofloxacin has areas of “medium” and “high degree of loading” (Figure 6b). Eugenol competes with levofloxacin when interacting with a conjugate. However, due to the fact that it is in a different phase, the inclusion degree of eugenol is much less (Figure 6c). However, in the regions from yellow and above to pink on the scale, double Lev–EG–conjugate complexes were obtained. Moreover, it is assumed that a very small amount of an adjuvant, for example, eugenol [46,47,48], presumably acting as a synergist for mechanisms that are being studied, such as blocking mitochondrial dehydrogenase, is required to enhance the antibacterial effect of fluoroquinolones. Thus, synthesized molecular containers with a combined drug are promising in the treatment of a wide range of diseases.

The interaction of levofloxacin with monomeric CD (as a control sample for comparison) and to a greater extent with its conjugates leads to significant changes in the environment of the aromatic system and functional groups (Appendix A). Shifts of characteristic peaks of Lev to the high-frequency region are observed: the peak of carbonyl C=O shifts from 1616 cm^−1^ in simple Lev to 1620 and 1622 cm^−1^ in complexes with CD and HPCD-PEI35-Man-**5** conjugate, respectively. At the same time, the maximum in the spectra of Lev shifts less—up to 1619 cm^−1^. It is known that Lev is included in the CD’s cavity from the side of the carboxyl group [43,70], but only in a protonated form, which is accompanied by the disappearance of a peak at 1580 cm^−1^ corresponding to COO^–^ due to protonation and immersion in the top of the CD [43,70]. We have discovered the fact that the aromatic peak (C–C 1474 cm^−1^) in solid-phase spectra is very sensitive to the microenvironment of levofloxacin, with a separation into two components. Similar results were observed earlier for moxifloxacin– HPCD inclusion complex [72]. In our case, peaks at 1452 cm^−1^ and 1492 cm^−1^ indicate two modes of complexation process between Lev and HPCD-PEI35-Man-**5**—thus, conjugates loaded with a drug were obtained. In the IR spectra of dissolved substances (Figure 5a and Appendix A), such splitting clearly does not occur; however, the appearance of shoulder peaks is noticeable, and deconvolution by Gaussians into the same 2 components is possible [72]—Appendix A—it is characterized by an increase in the proportion of the component at 1473 cm^−1^. Thus, solid-phase spectra complement the spectra of solutions since they exclude hydration.

The dissolution of eugenol in a polymer conjugate proceeds much more efficiently than with a simple CD, and also leads to changes in the IR spectrum (Appendix A), especially in the aromatic peak at 1515 cm^−1^. The presence of an alkyl-aryl ether peak at 1270 cm^−1^ confirms the presence of eugenol in the double complex.

#### 2.3.4. Kinetics of Drug Release from Molecular Containers

Promising approaches to the treatment of diseases, where the driver or direct participants are macrophages, imply targeted and prolonged release of drugs to reduce the dose and frequency of taking the drug. Simple cyclodextrins form inclusion complexes with fluoroquinolones with constants 10^3^–10^4^ M [72,73,74], which does not lead to a noticeable effect of prolonged release [75]. Therefore, molecular systems are being developed that can retain drugs with greater affinity, for example, polymer meshes of cross-linked cyclodextrins that increase the effective release of the drug up to 1–5 days [40]. We have synthesized conjugates **1–9** of different molecular architectures and, consequently, with different properties, including drug release parameters. We used levofloxacin and eugenol to study the kinetics of dissociation of its complexes with molecular containers. Figure 7 shows the kinetic curves of the levofloxacin and EG release in free form through a dialysis membrane (12–14 kDa cut-off) into an external solution in comparison with linear conjugate **5** (PEI35 grafted with cyclodextrins), star-shaped conjugates **7–8** based on spermine or light PEI1.8 and CD, as well as CD with branched PEI10 (conjugate **9**) due to the dissociation of the complexes.

Table 4 shows the kinetic parameters of Lev and EG release under the same conditions. Molecular containers provide approximately twice orthrice the slow half-release of drugs. Using the asymptotes, we determined the approximate content of levofloxacin and eugenol in the CD tori and retained by the oligomeric/polymer matrix (Table 4). Levofloxacin, in comparison with eugenol, is included in the cyclodextrin cavity to a greater extent (higher affinity) than entangled in the matrix. Both levofloxacin and eugenol are characterized by a distribution in favor of cyclodextrin tori in the case of star-shaped conjugates, and in favor of a polymer chain for a linear grafted conjugate. Due to the time spent on dissociation of drug complexes with molecular containers, the half-release period can be increased by 2–3 times (Table 4). And the total effective time of action increases significantly from half an hour to tens of hours. The slowest release of Levofloxacin from conjugates is observed for conjugates of **7**, **9**—star CD with grafted chains of spermine or PEI10. For eugenol, the slowest release is observed for conjugates of **8–9**—star CD with grafted chains of PEI1,8 and PEI10. These data correlate with the values of dissociation constants in comparison with monomeric HPCD (10^−3^ M) (Table 3).

Thus, we have obtained molecular containers that can be adapted directly to the tasks of a specific therapeutic strategy—by regulating the composition of the conjugate (components and their ratio), as well as the molecular architecture.

### 2.4. Antibacterial Activity of Levofloxacin and Adjuvants in Molecular Containers

The antibacterial activity of free levofloxacin and in the form of inclusion complexes with molecular containers HPCD-PEI35-Man-**5** and HPCD-spermine-Man-**7** was studied by the method of diffusion in agar containing *E. coli* cells (Table 5). The drug in the composition of conjugates has an antibacterial effect slightly superior in strength to free Lev, probably due to enhanced penetration through the cell membrane [40] and due to increased adhesion of conjugates to the cell wall [37].

Adjuvants (allylpolyalkoxybenzenes ) can be used to reduce the concentration of the toxic main component of levofloxacin in the complex formulation due to the synergy effect, as well as to increase the therapeutic potential of the drug. Presumably, the use of complex forms of Lev in combination with allylpolyalkoxybenzenes would increase the circulation time of Lev in the bloodstream and “save” a significant part of the drug from misuse (destruction, accelerated excretion). Indeed, while eugenol has an antibacterial effect on some bacteria [73,76], it is found that eugenol [74], apiol and its derivatives [77,78], and some monoterpenes such as menthol and linalool [79] showed synergism with several antibiotics. 

The introduction of the synergist eugenol into conjugates further enhances the effect of levofloxacin up to 15% (Table 5). Thus, the minimum inhibitory Lev concentration can be reduced from 0.4–0.45 µg/mL to 0.2–0.35 µg/mL due to both the molecular container and adjuvants. However, this effect is not the key.

The most significant effect is that the molecular containers provide a prolonged effect of levofloxacin (Figure 8). Bacterial growth was suppressed by all samples with Lev for 24 h. However, based on the steepness of the initial slope of curves (Figure 8b) and absorption of cell suspension (A_600_ values) after 22 h (Figure 8b), it can be concluded that EG (eugenol) and apiol enhance and accelerate the antibacterial effect of Lev in conjugates (with comparably high efficiency in both HPCD-spermine-Man-**7** and HPCD-PEI35-Man-**5**), and therefore the MIC of the antibiotic can potentially be reduced. Statistical analysis of obtained data was carried out using the Student’s t-test show that the differences in the Lev efficiency in free form and in the complex with conjugates (values of both groups of experiments are presented in Table 5) were considered statistically significant, since the calculated *p*-value was less than 0.05. Comparing the values of absorption of cell suspension (correlates with the number of colony-forming units—CFU) after 1 and 6 days, we come to the conclusion that the Lev in the complex with conjugates acts more than 50% more effectively than the free form. Moreover, the effect of prolonged action is observed (the decline of curves against the plateau in the case of Lev). In this article, we have demonstrated that eugenol and apiol exhibit a synergy effect with Levofloxacin, boosting its antibacterial activity at concentrations several orders of magnitude lower than is needed for these substances to show any antibacterial activity. The ratio of Levofloxacin:eugenol was 1:1 and 1:2. Here we aimed at showing the proof-of-principle—that there is a possibility as such to obtain the double complexes of Levofloxacin and its adjuvants (allylpolyalkoxybenzenes), and to study the ligand binding constants, release kinetics and the antibacterial activity of drug molecules from both single and double complexes. 

In another paper (in press), we found the ratio of Levofloxacin:eugenol = 1:500—1:1000 to be optimal, yielding antibacterial activity of Levofloxacin at concentrations several fold lower than would normally be expected [48]. This synergistic effect can be explained by the formation of defects in the bacterial membrane by an aromatic molecule, resulting the increased penetration of the antibiotic into the cell; and also, by the inhibition of bacterial outflow pumps and, as a consequence, increase of intracellular antibiotic concentration. 

Thus, the combined formulation of levofloxacin-adjuvant-cyclodextrin can be considered as a promising drug system for the treatment of bacterial infections in order to increase the therapeutic effect, minimize side effects on the body and ensure the absence of resistance of pathogenic microorganisms.

## 3. Materials and Methods

### 3.1. Materials

We used carbonyldiimidazole—CDI (Chemical Process, Singapore), PEI 30–40 kDa (Serva, Heidelberg, Germany), ConA (Paneco-ltd, Moscow, Russia). Spermine, D-mannose, PEI1.8 and PEI10, fluorescein isothiocyanate (FITC), DMF, DMSO, Et_3_N, orthophthalic aldehyde, Et_2_O, 2-mercaptoethanol and 2-hydroxipropyl-β-cyclodextrin were obtained from Sigma Aldrich (St. Louis, MI, USA), levofloxacin from Zhejiang Kangyu Pharm Co Ltd. (Zhejiang, China). Other chemicals: 2,4,6-trinitrobenzenesulfonic acid, NaBH_4_, salts and acids—production Reakhim (Moscow, Russia).

### 3.2. Synthesis of HPCD Derivatives with PEI and Spermine

Adapted methodology [52,53,54,55,56] used for obtaining mesh and linear polymer conjugates: 6 mixtures of dry substances were prepared, consisting of HPCD sample (0.038 g, 25 mmol) and an excess of CDI (10-, 20- and 40-fold molar surpluses—two samples each). The powders are dissolved in 0.5 mL of DMF. The samples were incubated for an hour at room temperature. Then 1.5 mL of cooled to −70 °C Et_2_O was added to precipitate a transparent gel (HPCD-CDI), which was separated for further synthesis. Three solutions of spermine for conjugates 1–3 and PEI (35 kDa) for conjugates 4–6 were prepared in 50% DMSO with a concentration of 0.1 g/mL. The molar excesses relative to the initial amount of HPCD are respectively 8, 15 and 30 for spermine; 0.25, 0.04 and 1 for PEI. HPCD-CDI gels and 0.1 mL Et_3_N were added drop-by-drop to the indicated solutions of spermine and PEI for 90 min. The samples were incubated for 6 h at room temperature. In solutions 1–3 containing spermine, a white precipitate (22 mg) was formed, which was separated by centrifugation and dissolved in 1 mL of water. Purification of the obtained six masterbatch solutions of the samples (HPCD-PEI and HPCD-spermine conjugates) was carried out in 5–8 consecutive stages of adding 500–1000 µL of water and centrifugation with filters of 3 kDa in the case of spermine and 30 kDa in the case of PEI (5000 rpm, 30 min, Eppendorf).

Other techniques were used for obtaining star and branching polymer conjugates, as well as increasing the quantitative yields of intermediates and products, and also for narrowing the distribution by structure. The slow drop-by-drop addition of HPCD solution in water (0.4 mM) to the CDI solution in DMSO (1.5 mM) makes it possible to create a locally very high excess of the activating agent and thereby obtain a product of a strict HPCD-CDI_7_ composition at the output. Activated cyclodextrin was added drop by drop to a solution of PEI or spermine. Purification from unreacted components and low molecular weight substances was carried out by dialysis (cut-off weight from 3.5 to 14 kDa, depending on the type of oligo- or polyamine). The technique does not require the use of ice ether for precipitation, and the amount of organic solvents is reduced, which simplifies the purification stages and simplifies the technological implementation of pharmaceutical production of these delivery systems at the enterprise. Schemes of synthesis are given in (Appendix A).

### 3.3. Mannosylation and Purification of HPCD-PEI and HPCD-Spermine Conjugates

To samples of 200–600 µL of HPCD-spermine and HPCD-PEI conjugates with an average concentration of 25 mg/mL and 60–200 mg/mL respectively, a one-and-a-half-fold molar excess (relative to the number of NH_2_-groups) of 3 M mannose water solution was added drop by drop with intensive stirring for 1 h, followed by small portions of dry Schiff base reducing agent NaBH_4_. Purification of conjugates from low molecular weight impurities was carried out using centrifuge filters, as described above. The purity of the preparation was controlled by HPLC gel filtration in a Knauer chromatography system (Knauer, Berlin, Germany) on BioFox 17 SEC in a 15 cm × 1 cm^2^ column. The eluent was 15 mM PBS (pH 7.4) containing 150 mM NaCl; the elution rate was 0.5 mL/min, 25 °C. The chromatogram of the resulting conjugate is shown in Appendix A. The resulting conjugates were lyophilized or frozen and stored at −20 °C.

The degree of mannosylation was calculated according to spectrophotometric titration of amino groups (before and after mannosylation) with 2,4,6-trinitrobenzenesulfonic acid [28] and fluorescence analysis with orthophthalic aldehyde and 2-mercaptoethanol [80,81] (Appendix A). Mainly primary amino groups are detected. 

### 3.4. FTIR-Spectroscopy

ATR-FTIR spectra of samples’ solutions were recorded using a Bruker Tensor 27 spectrometer equipped with a liquid nitrogen cooled MCT (mercury cadmium telluride) detector. Samples were placed in a thermostated cell BioATR-II with ZnSe ATR element (Bruker, Germany). The IR spectrometer was purged with a constant flow of dry air (Jun-Air, USA). FTIR spectra were acquired from 900 to 3000 cm^−1^ with 1 cm^−1^ spectral resolution. For each spectrum, 50–70 scans were accumulated at 20 kHz scanning speed and averaged. ATR-FTIR spectra of solid samples placed on KBr glass were recorded using a Bruker Lumos II IR microscope in region from 700 to 4000 cm^−1^ with 1 cm^−1^ spectral resolution with scanning in the area on average 1 × 1 microns. Spectral data were processed using the Bruker software system Opus 8.2.28 (Bruker, Germany), which includes linear blank subtraction, baseline correction, differentiation (second order, 17 smoothing points), min-max normalization and atmosphere compensation [43,82]. If necessary, 5-point Savitsky–Golay smoothing was used to remove white noise. Peaks were identified by standard Bruker picking-peak procedure.

#### 3.4.1. Complex Formation of ConA with Conjugates. FTIR

Titration of a dimeric ConA solution (C = 33 µM) in 25 µL of a sodium citrate buffer solution (C = 0.02 M, pH 5.5) containing 0.5 M NaCl, 1 mM CaCl_2_, 1 mM MnCl_2_ was carried out by adding aliquot ligands (HPCD-PEI-Man and HPCD-spermine-Man conjugates) 0.5–50 µL at 22 °C. The degree of complexation was determined by changes in the intensity of amide II band normalized to the amide I band, or the same changes of intensity or max-min position in Amide II areas in differential IR spectra of the second order.

#### 3.4.2. Complex Formation of Conjugates with Levofloxacin. FTIR

Levofloxacin complexes (0.67–5 mg/mL) with HPCD-PEI-Man or HPCD-spermine-Man conjugates in 25–150 µL diluted HCl solution (C_HCl_ = 1 mM, pH = 3) were first prepared by mixing samples and one hour incubation at 37 °C. The concentration per CD ring varied in the range 10 μM–10 mM) at 37 °C. Further, the IR spectra of the samples were recorded. The proportion of bound levofloxacin was determined by the intensity of the aromatic peak at 1474 cm^−1^, as well as the position of the minimum-maximum of the second derivative in this region.

#### 3.4.3. Complex Formation of Conjugates with Eugenol. FTIR

Eugenol complexes (0.43–2.4 mg/mL-aqueous suspension, pre-moistened eugenol oil of 10 µL ethanol) with HPCD-PEI-Man or HPCD-spermine-Man conjugates were prepared by mixing samples and one hour incubation at 37 °C. Further—similarly as in the case of levofloxacin. Analytical aromatic peak at 1515 cm^−1^.

#### 3.4.4. Formation of Double Drug Inclusion Complexes of Eugenol and Levofloxacin

Solid-phase (heterogeneous) synthesis. Two samples of HPCD-PEI35-Man-5 with Lev and a star-shaped conjugate HPCD-spermine-Man-7 with Lev were obtained, moistened with eugenol oil (molar ratio CD tori: Lev: EG = 1:1:1), and a couple of drops of ethanol or acetonitrile were added. The sample is ground on glass and dried at low heat for no more than 35 °C for 30 min. Then leave at room temperature.

### 3.5. Fluorescence Spectroscopy

Fluorescence was measured using a Varian Cary Eclipse spectrophotometer (Agilent Technologies, Santa Clara, CA, USA) at a temperature of 22 °C in the fluorescence range of Trp residues (for nonfluorescent ligand λ_ex_ = 280 nm, λ_em_ = 337 nm) and FITC-labeled ligand (λ_ex_ = 485 nm, λ_em_ = 525 nm) [18]. Fluorescence anisotropy spectra were recorded using a manual polarizer. The fluorescence anisotropy value was calculated based on the intensity components according to the formulas:(1)$$r=\frac{I_{VV}-G\cdot I_{VH}}{I_{VV}+2G\cdot I_{VH}},$$
(2)$$\mathrm{factor}\;G=\frac{I_{HV}}{I_{HH}},$$

#### 3.5.1. Complex Formation of ConA with Conjugates. Fluorescence Methods

Titration of a dimeric ConA solution (C = 19 µM) in 500 µL of a sodium citrate buffer solution (C = 0.02 M, pH 5.5) containing 0.5 M NaCl, 1 mM CaCl_2_, 1 mM MnCl_2_ was carried out by adding aliquot ligands (HPCD-PEI-Man and HPCD-spermine-Man conjugates) 2–400 µL at 22 °C. The degree of complexation was determined by changes in the intensity and anisotropy of the fluorescence of tryptophan residues in the receptor protein.

#### 3.5.2. Complex Formation of ConA with FITC-Labeled Ligands. Fluorescence Methods

Titration of a FITC-labeled ligand solution (C_FITC_ ≈ 0.2 µM) in 550 µL Tris/HCl buffer solution (C = 0.01 M, pH = 7.2) containing 0.5 M NaCl, 1 mM CaCl_2_, 1 mM MnCl_2_ was performed by adding aliquot ConA (10–25 mg/mL) 2–200 µL at 22 °C. The degree of complexation was determined by changes in the intensity and anisotropy of FITC fluorescence.

#### 3.5.3. Complex Formation of Conjugates with Levofloxacin. Fluorescence Methods

Titration of a levofloxacin (135 µM) in 500 µL diluted HCl solution (C_HCl_ = 1 mM, pH = 3) was performed by adding aliquot HPCD-PEI-Man or HPCD-spermine-Man conjugates (the concentration per HPCD ring varied in the range 10–20 mM) at 37 °C. The degree of Lev inclusion was determined by changes in the intensity and anisotropy of Lev fluorescence (λ_ex_ = 396 nm, λ_em_ = 480 nm).

### 3.6. Dynamic Light Scattering (DLS)

The particle sizes and zeta potentials were measured using a Zetasizer Nano S «Malvern» (England) (4mW He–Ne-laser, 633 nm, scattering angle 173°). The experiment was performed in a temperature-controlled cell at 25 °C. Autocorrelation functions of intensity fluctuations of light scattering were obtained using the correlation of the Correlator system K7032-09 «Malvern» (Worcestershire, UK). Experimental data was processed using «Zetasizer Software» (v. 8.02).

### 3.7. Levofloxacin and Eugenol Release by UV-Spectroscopy 

Parameters of release kinetics of free levofloxacin (C_0_(Lev) = 0.2 mg/mL), eugenol (C_0_(EG) = 2 mg/mL) and eugenol/levofloxacin complexes with molecular containers **5**, **7–9** through a dialysis membrane (12–14 kDa cut-off) into an external solution (1 mL: 5 mL) were measured by UV-absorption at 287 (Lev) and 280 (EG) nm. Aliquots of 400 µL were selected and replaced with a buffer solution. UV-spectroscopy was performed with Amersham Biosciences Ultraspec 2100 pro. UV-spectra were recorded 240–400 nm at 25 °C. PBS (pH = 7.4).

### 3.8. NMR Spectroscopy

12 mg of the sample was dissolved in 700 µL of D_2_O. ^1^H -spectra of the solutions were recorded on a Bruker Avance 400 spectrometer (Germany) with an operating frequency of 400 MHz. Chemical shifts (δ) in m.d. were calculated relative to the residual signals of D_2_O (4.79 m.d.).

### 3.9. Antibacterial Activity of Levofloxacin and Adjuvants in Molecular Containers

The strains used in this study were *Escherichia coli* (NCIB 12210) from National Resource Center All-Russian collection of industrial microorganisms SIC “Kurchatov Institute”). The cell sample was placed in 5 mL of liquid medium Luria–Bertani (pH 7.2) in a thermostat at 37 °C. 24 h before the experiments, 150 µL of cell culture was placed in 3 mL of liquid nutrient medium Luria–Bertani (pH 7.2) and then in a thermostat at 37 °C without stirring. The study of the antibacterial effect of levofloxacin and adjuvants was carried out by agar diffusion test. 500 µL of 100-fold culture dilution was evenly distributed over a solid nutrient medium (Luria–Bertani, pH 7.2) on a Petri dish. After 20 min, agar disks with a diameter of 9 mm were removed from the cups and 50 µL of the test samples were placed in the formed wells at the required concentration in a sterile PBS buffer (see paragraph 3.3(2)). After 30 min, Petri dishes were placed in a 37 °C thermostat. After 24 h, the diameters of the growth inhibition zones were measured, and the minimum inhibitory concentrations and amounts of adjuvants necessary to enhance the main component of levofloxacin were calculated.

The study of the antibacterial effect of fluoroquinolones in liquid media was carried out using a nocturnal bacterial culture. The cell suspension was diluted twice with Luria–Bertani liquid medium (pH 7.4). 200 µL of sample was added to 1.8 mL of culture. The resulting systems were incubated at a temperature of 37 °C and stirring at 100 rpm in an ES-20 BIOSAN incubator shaker (Latvia) for 6 days. Aliquots were selected at certain intervals to measure absorption at 600 nm—to determine the number of viable cells.

### 3.10. Mathematical Processing of the Results


(a)Calculation of dissociation constants of ConA-ligand complexes was carried out in 3 stages:
(1)Fitting of the curves of change of the analytical signal ξ versus molar excess of the ligand x was carried out using the Hill and Boltzmann equations (Origin software):
Hill1: ξ = start + (end–start) · x^n^/(x^n^ + k^n^)(3)
where start and end—horizontal asymptotes;
Boltzmann: ξ = (A_1_–A_2_)/(1 + exp((x−x_0_)/dx)) + A_2_(4)(2)Calculation of the fraction of the bound receptor (α) through the values of the analyzed quantities (fluorescence intensity (F) or anisotropy (r), IR absorption (A)) for the receptor ξ_0_, the complex ξ_∞_ and the current value ξ, as described earlier [18]:
(5)$$\alpha =\left|\frac{\xi \;-\;\xi_{0}}{\xi_{\infty }\;-\;\xi_{0}}\right|,$$(3)Calculation of the equilibrium concentrations of the receptor [R], ligand [L] and complex [R·L] and the *K*_d_ value by following equations by following equations [63,83]:
material balance for receptor: C_0_(R) = [R] + [R·L](6)
material balance for ligand: C_0_(ligand) = [L] + [R·L](7)
complex: [R·L] = C_0_(R) ∙ α(8)
(9)$$K_{d}=\frac{\left[R\right]\;\cdot \;\left[L\right]}{\left[R\cdot L\right]}—\mathrm{one}-\mathrm{binding}\;\mathrm{site}\;\mathrm{model}$$
Hill’s linearization model: lg (θ/(1–θ)) = n · lg [L]—lg *K*_d_,(10)
where θ = [R·L]/C_0_(R)—proportion of bound receptor, n—number of binding site.
Scatchard’s linearization model: [R·L]/[L] = (C_0_(R)—[R·L])/*K*_d_(11)


Similar calculations were performed for the Lev and EG—receptor interactions.


(b)Determination of drug loading parameters:(1)Entrapment efficiency of X [50]: EE (%) = 100·(X amount of loaded X)/(total X amount).(2)Loading capacity of X [84]: LC(%) = 100·(mass of loaded X)/(mass of sample) = 100 · (number of drug molecules per one container molecule) · (molar mass of X)/(molar mass of conjugate).


Statistical analysis of obtained data was carried out using the Student’s *t*-test and Statistica 9.0 software (StatSoft, Tulsa, OK, USA).

## 4. Conclusions

In this paper, the spectral approach for high-throughput screening of lectin-ligand interactions using concanavalin A as a model mannose receptor in application for molecular containers based on cyclodextrins, oligoamines and mannose for targeted delivery of levofloxacin to alveolar macrophages have been developed to increase the effectiveness of therapy for various diseases, including pneumonia and tuberculosis. By varying the conjugate structure from mesh, polymer to star-shaped and branched, it is possible to adapt the delivery system to specific treatment tasks: the time of drug half-release and effective action. The introduction of a mannose label (instead of using expensive derivatives of oligomannosides) allows targeting only activated macrophages expressing the CD206 mannose receptor, which minimizes adverse reactions and immune response. Complexes of inclusion of aromatic model fluoroquinolone levofloxacin into cyclodextrin tori and polymer matrix have been synthesized, as well as promising adjuvants eugenol and apiol—antibacterial enhancers. Conjugate **7** (star-shaped cyclodextrin with grafted spermines, respectively) demonstrate the highest parameters of loading capacity and entrapment efficiency to both levofloxacin and eugenol molecules parameters (EE and LC). These conjugates combine the valuable properties of a drug container and a high affinity ligand to mannose receptors due to presence of the trimannoside analogue on spacer spermine. Thus, the developed systems of targeted drug delivery of combined drugs due to their high specificity to macrophage mannose receptors (shown in the model ConA), presumably low immunogenicity, high drug loading capacity and prolonged release in target tissues are promising in medical biotechnology and pharmaceuticals for the creation of new dosage forms and strategies for the treatment of respiratory tract diseases.

## Figures and Tables

**Figure 1 pharmaceuticals-15-00625-f001:**
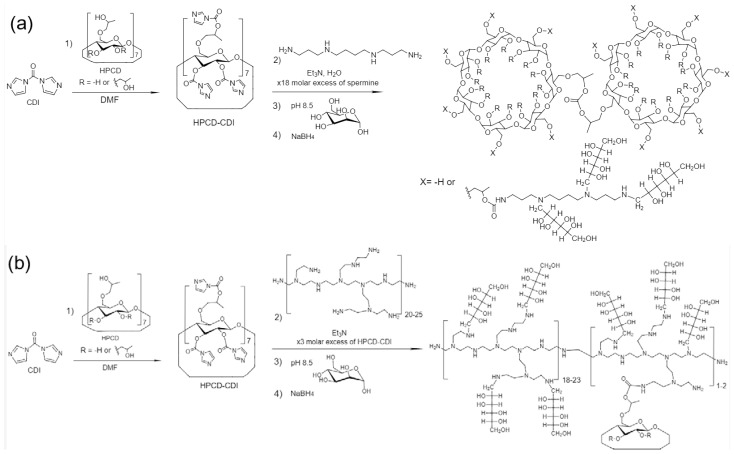
(**a**) The scheme of synthesis of mannosylated star-shaped HPCD-spermine-Man-**7** conjugate. The characteristics of the synthesized ligand are given in Table 1. The number of substituents X, not hydrogen atoms, is on average 9. (**b**) The scheme of synthesis of mannosylated HPCD-PEI10-Man-**9** conjugate. The characteristics of the synthesized ligand are given in Table 1. On average, one branched mannosylated chain of PEI accounts for 1–2 cyclodextrins.

**Figure 2 pharmaceuticals-15-00625-f002:**
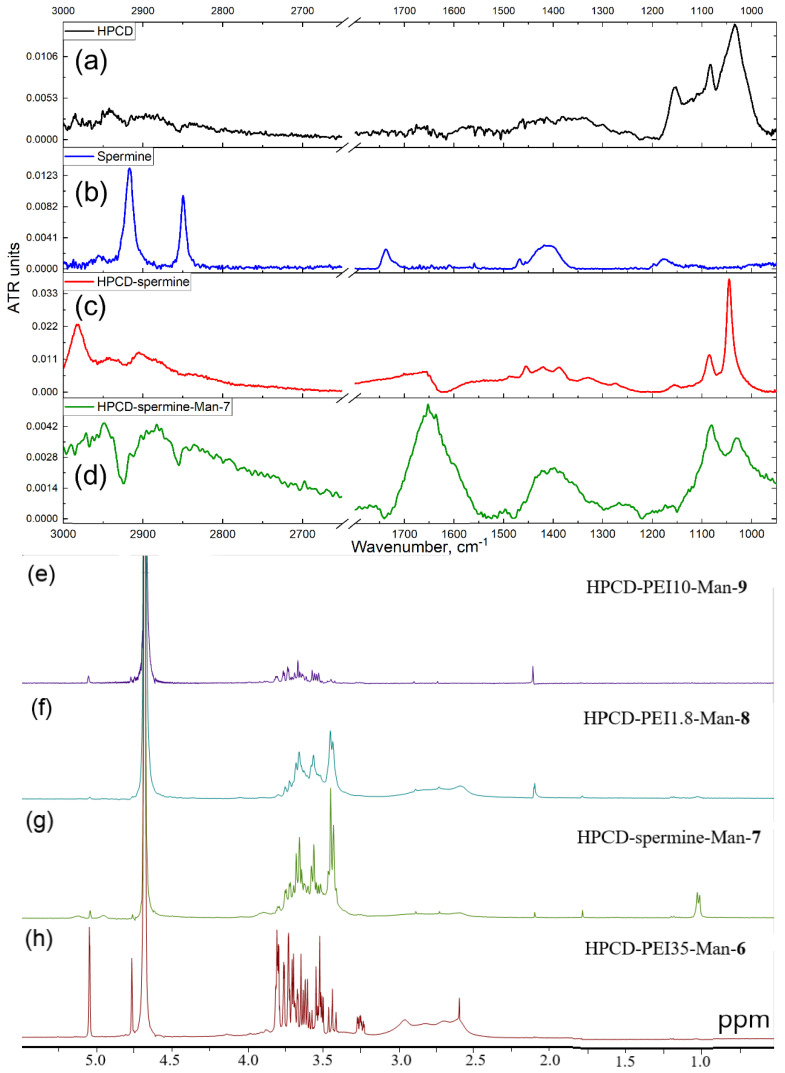
Fourier transform infrared spectra of (**a**) HPCD, (**b**) spermine, (**c**) HPCD-spermine conjugate, (**d**) mannosylated HPCD-spermine-Man-**7.**
^1^H NMR spectra of conjugates: (**e**) HPCD-PEI10-Man-**9**, (**f**) HPCD-PEI1.8-Man-**8**, (**g**) HPCD-spermine-Man-**7**, (**h**) HPCD-PEI35-Man-**6**. D_2_O, 12 mg/mL 400 MHz.

**Figure 3 pharmaceuticals-15-00625-f003:**
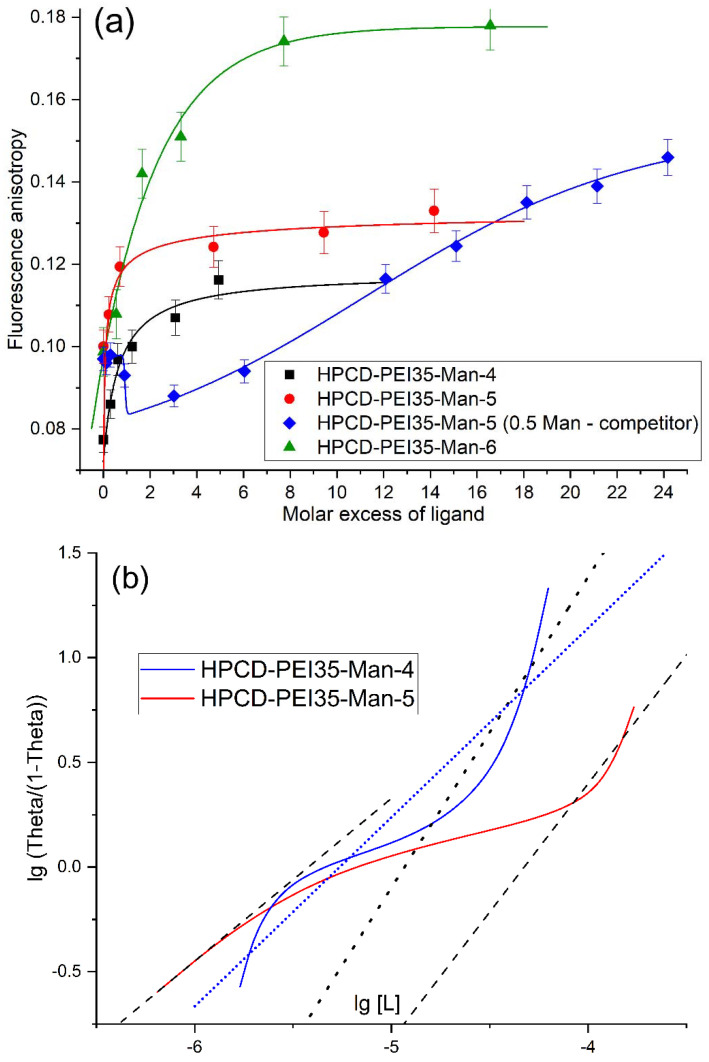
(**a**) Tryptophan fluorescence anisotropy plot: titration of ConA by ligands HPCD-PEI35-Man group. Fitted with Hill (3), Boltzmann (4) and BiDoseResp equations from Origin. (**b**) Hill’s linearization plot of fluorescence anisotropy. [L]—the equilibrium concentration of the ligand. Theta-proportion of bound receptor. Two dash lines (– –) for red curve are a linearized model of two types of binding. Two dot lines (‘‘‘) for blue curve are a linearized model of the most affinity binding and average observed binding (short dots). C_0_(dimeric ConA) = 38 μM. Ligand concentration varied from 1 to 250 μM. In the case of competitor binding (Appendix A), we used mannose with [Man] = 0.5 M. Natrium-citric buffer solution (0.04 M, pH 5.5). C(Ca^2+^) = C(Mn^2+^) = 1 mM. T = 22 °C.

**Figure 4 pharmaceuticals-15-00625-f004:**
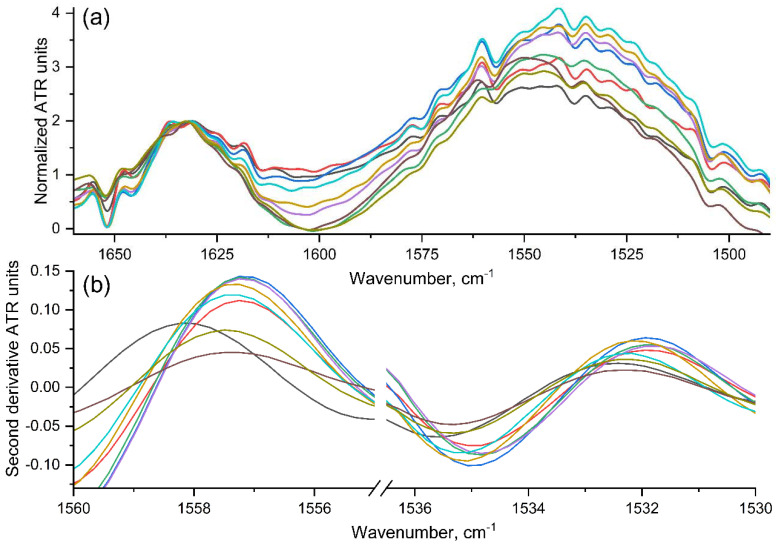
(**a**) Normalized to the amide I band Fourier transform infrared spectra of ConA: titration of ConA by ligand HPCD-spermine-Man-**1**. The sequential addition of the ligand corresponds to the transition from blue to red and black, i.e., from top to bottom. (**b**) Amide II areas in differential IR spectra of the second order. C_0_(ConA) = 33 μM. Ligand concentration varied from 90 to 750 μM. Natrium-citric buffer solution (0.04 M, pH 5.5). C(Ca^2+^) = C(Mn^2+^) = 1 mM. T = 22 °C.

**Figure 5 pharmaceuticals-15-00625-f005:**
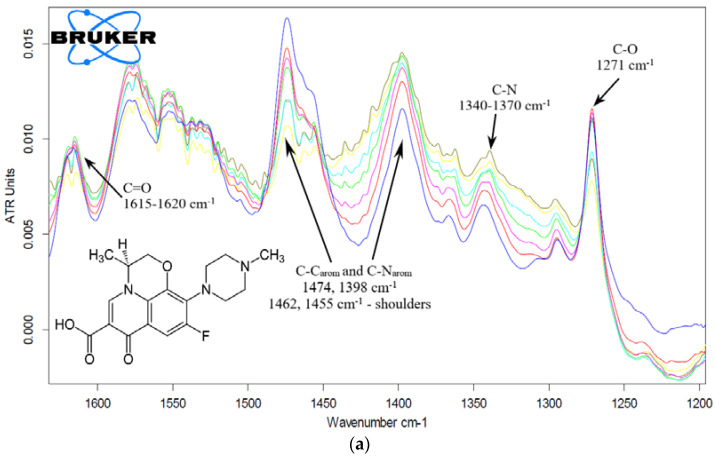
(**a**) Fourier transform infrared spectra of levofloxacin complexes with HPCD-spermine-Man-**1**. C_0_(Lev) = 5 mg/mL. 1 mM HCl solution. Molar ratio HPCD tori of ligand:Lev is 0, 0.06, 0.11, 0.23, 0.4, 0.6, 0.93 respectively for gradient spectra from blue to brown. (**b**) FTIR spectra of eugenol complexes with HPCD-PEI35-Man-**5** conjugate. C_0_(eugenol in water suspension) = 0.43 mg/mL. The concentration of the ligand per cyclodextrin tori ranged from 8 to 260 μM. T = 22 °C.

**Figure 6 pharmaceuticals-15-00625-f006:**
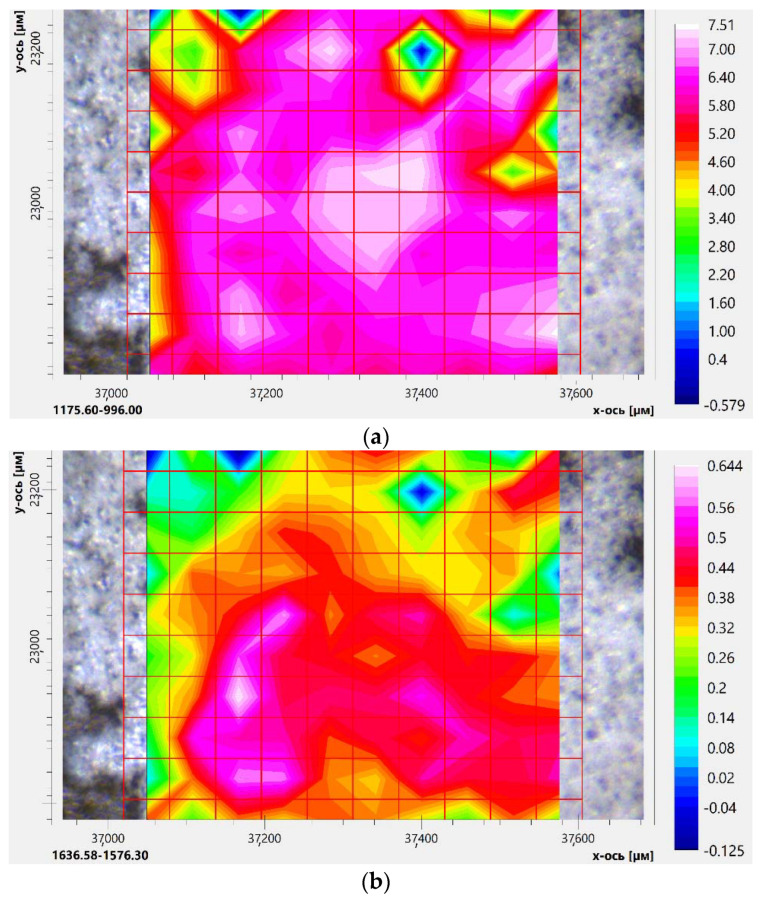
Integral intensity of three regions in solid phase FTIR spectra of levofloxacin (Lev) and eugenol (EG) double complex with HPCD-spermine-Man-**7** conjugate, superimposed on microphotography samples of 1.5 × 0.5 microns in size. (**a**) HPCD region—996–1175 cm^–1^. (**b**) Levofloxacin region—1576–1636 cm^–1^. (**c**) Eugenol region—1502–1522 cm^–1^. (**d**) Solid phase FTIR spectra in 7 points (the same substance and region).

**Figure 7 pharmaceuticals-15-00625-f007:**
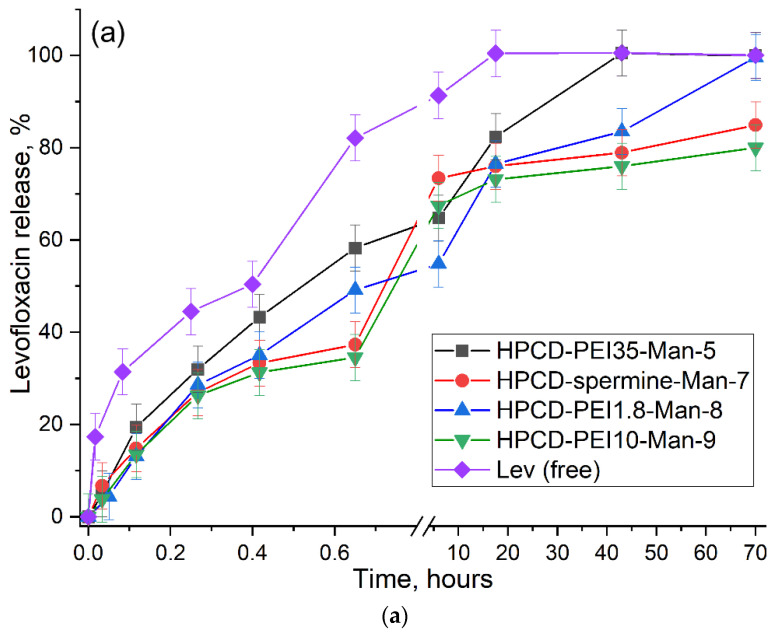
Release kinetics curves of (**a**) free levofloxacin and levofloxacin complexes (1:1 by mass) with molecular containers **5**, **7–9**; and (**b**) free eugenol and eugenol complexes (1:1 by mass) with molecular containers **5**, **7–9**. Dialysis membrane (12–14 kDa cut-off) into an external solution (1:5 by volumes). PBS (pH = 7.4). Levofloxacin and eugenol are detected by UV-absorption at 287 and 280 nm, correspondingly. C_0_(Lev) = 0.2 mg/mL. C_0_(eugenol) = 2 mg/mL. T = 37 °C. Values are presented as the mean ± SD of three experiments.

**Figure 8 pharmaceuticals-15-00625-f008:**
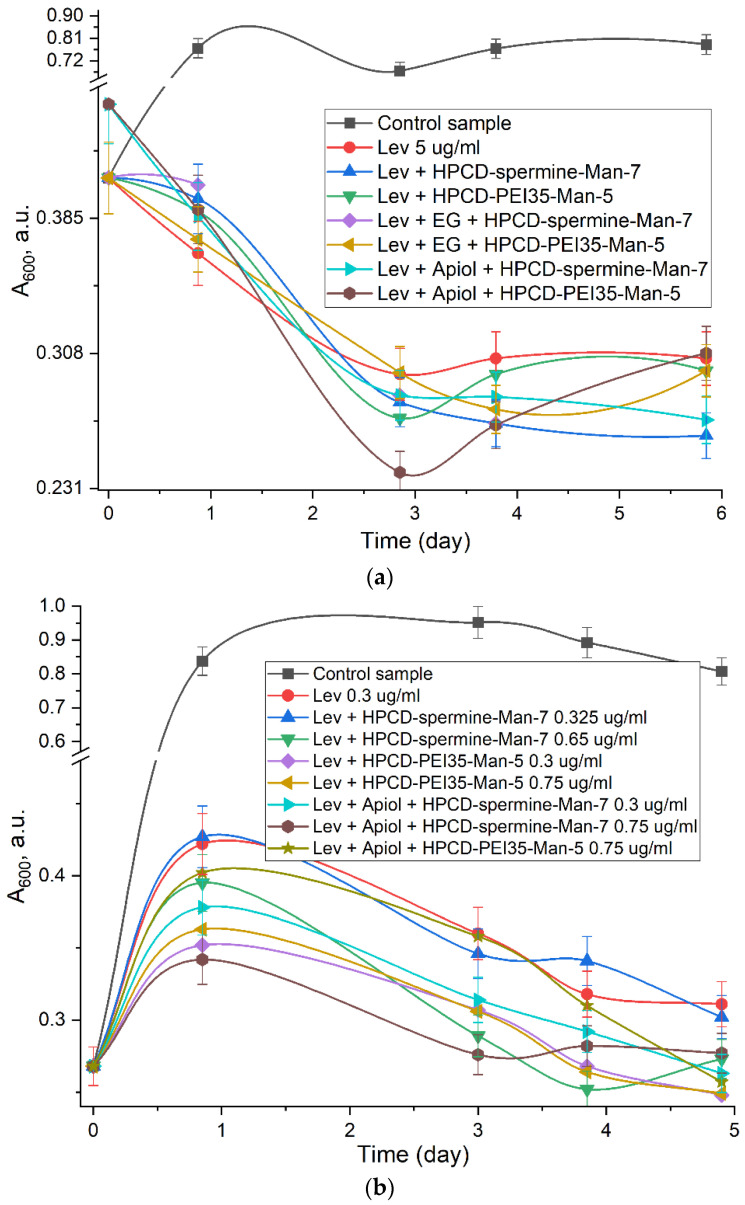
The dependence of the optical density A_600_ (correlate with CFU) on the incubation time of *E. coli* cells (NCIB 12210) with drugs. (**a**) In all samples C(Lev) = 5 μg/mL (**b**) C(Lev) = 0.3–0.75 μg/mL are indicated in the figure. Molar ratio of Lev:apiol and Lev:EG is equal to 1:1. pH 7.4 (0.01 M PBS), 37 °C, 24 h of incubation. Values are presented as the mean ± SD of three experiments.

**Table 1 pharmaceuticals-15-00625-t001:** Physical-chemical properties of obtained β-cyclodextrin conjugates.

Sample’s Designation *	Hydrodynamic Diameter (by DLS) **, nm	M_r_, kDa	Molar Ratio HPCD: Spermine (or PEI): for both levofloxacin and eugenol.Man	Average Amount of Mannose Residues per Molecule	Degree of Mannosylation	ζ-Potential **, mV	References to Synthesis Schemes and FTIR/NMR Spectra
HPCD-spermine-Man-**1**	336 ± 68	6	7.4:1:6.2	6	95 ± 3	−2.7 ± 0.8	Appendix A
HPCD-spermine-Man-**2**	523 ± 106	9	5:1:6.9	7	98 ± 2	−4.1 ± 1.9	Appendix A
HPCD-spermine-Man-**3**	781 ± 205	8.5	1.5:1:11	22	92 ± 3	−4.9 ± 0.1	Appendix A
HPCD-PEI35-Man-**4**	52.8 ± 1.3	42	1:1:34.5	35	77 ± 3	5.1 ± 1.4	Appendix A
HPCD-PEI35-Man-**5**	62.3 ± 0.7	50	8.6:1:83	83	94 ± 4	−9.4 ± 2.1	Appendix A
HPCD-PEI35-Man-**6**	59 ± 7	115	1:2:271	271	80 ± 4	19.0 ± 2.5	Figure 2h, Appendix A
HPCD-spermine-Man-**7**	152.2 ± 1.0	8	2:9:21	21	85 ± 5	−6.0 ± 0.4	Figure 1a and Figure 2
HPCD-PEI1.8-Man-**8**	66.9 ± 0.3	20	1:6.1:45	45	83 ± 4	−2.0 ± 0.1	Figure 2f, Appendix A
HPCD-PEI10-Man-**9**	373 ± 108	28	2:1:154	154	99 ± 1	4.8 ± 1.0	Figure 1b, Figure 2e, Appendix A

* The last number in the name is the serial number of the sample, and the number after PEI is the average M_n_ molar mass of the polymer. ** SD for *n* = 3 is given.

**Table 2 pharmaceuticals-15-00625-t002:** Dissociation constants of ConA–ligand complexes determined by fluorescence methods and FTIR spectroscopy. The conditions are given in the Methods section.

Ligand	*K*_d_ (ConA–Ligand), μM			
	Quenching of Trp Fluorescence		FTIR Spectroscopy	
HPCD-spermine-Man-**1**	64 ± 11		50 ± 6	
HPCD-spermine-Man-**2**	69 ± 12		60 ± 11	
HPCD-spermine-Man-**3**	0.55 ± 0.19		0.49 ± 0.23	
HPCD-spermine-Man-**7**	0.24 ± 0.12		0.27 ± 0.08	
Ligand	Quenching of Trp fluorescence	Anisotropy of Trp fluorescence		FTIR spectroscopy
HPCD-PEI35-Man-**4**	11 ± 6	*K*_obs_ =17 ± 8		12 ± 3
HPCD-PEI35-Man-**5**	69 ± 21	*K*_obs_ = 60 ± 19 *K*_max_ = 5.0 ± 0.7		57 ± 17
HPCD-PEI35-Man-**6**	*K*_obs_ = 31 ± 12 *K*_max_ = 0.10 ± 0.03	*K*_obs_ = 1.3 ± 0.4 *K*_max_ = 0.97 ± 0.22		2.8 ± 0.7
HPCD-PEI1.8-Man-**8**	36 ± 13	*K*_obs_ = 19 ± 6 *K*_max_ = 2.9 ± 1.2		27 ± 10
HPCD-PEI10-Man-**9**	6.8 ± 0.4	8.1 ± 1.5		55 ± 14 *
Mannan (control)	1.9 ± 0.5	1.6 ± 0.9		1.7 ± 0.5

* Multiple site binding and positive cooperativeness. The value of the K_d,n_ dissociation constant in the Hill’s model is given. lg *K*_d_ = 13.71, *n* = 3.22. lg *K*_d,n_ = (lg K_d_)/*n*.

**Table 3 pharmaceuticals-15-00625-t003:** Dissociation and equilibrium constants of conjugates’ complexes with levofloxacin and eugenol determined by FTIR spectroscopy. Maximum loading capacity (N) of the drug into conjugates. Entrapment efficiency (EE) and loading capacity (LC) of levofloxacin and eugenol by molecular containers. The conditions are given in the Methods section.

	Levofloxacin					
Conjugate	HPCD-spermine-Man-**1**	HPCD-spermine-Man-**2**	HPCD-spermine-Man-**3**	HPCD-spermine-Man-**7**	HPCD-PEI35-Man-**5**	HPCD-PEI35-Man-**6**
–lg *K*_d_	3.89 ± 0.24	3.75 ± 0.17	3.33 ± 0.19	4.46 ± 0.29	5.28 ± 0.41	3.15 ± 0.15
N	14	19	12	15	85	80
EE, %	97 ± 2	96 ± 3	82 ± 3	98 ± 2	98 ± 2	93 ± 3
LC, %	86 ± 5	78 ± 4	52 ± 4	69 ± 4	63 ± 5	26 ± 2
	Eugenol					
Conjugate	HPCD-PEI35-Man-**5**		HPCD-PEI35-Man-**6**	HPCD-spermine-Man-**7**		
*K* *	47 ± 4		7 ± 2	58 ± 5		
N	6 ± 1 (mostly cyclodextrin inclusion)		7 ± 2 (mostly polymer’s interaction)	6 ± 1		
EE, %	83 ± 5		62 ± 6	86 ± 4		
LC, %	2.0 ± 0.1		1.0 ± 0.1	12 ± 1		

* *K* = [eugenol in complex]/C(cyclodextrin tori of conjugate).

**Table 4 pharmaceuticals-15-00625-t004:** Kinetics parameters of release curves of (a) free levofloxacin and levofloxacin complexes (1:1 by mass) with molecular containers **5**, **7–9**; and (b) free eugenol (EG) and eugenol complexes (1:1 by mass) with molecular containers **5**, **7–9**. Dialysis membrane (12–14 kDa cut-off) into an external solution (1:5 by volumes). PBS (pH = 7.4). Levofloxacin and eugenol are detected by UV-absorption at 287 and 280 nm, correspondingly. C_0_(Lev) = 0.2 mg/mL. C_0_(eugenol) = 2 mg/mL. T = 37 °C.

Conjugate (Sample)	Time of Semi-Release of Lev τ_1/2_, Min	Time of 80%–Release of Lev τ_80%_, Hours	Kinetic Constants, h^−1^		The Proportion of Lev or EG Included in Cyclodextrin Tori and Retained by the Polymer Matrix, %
			The Total Process of Dissociation of the Complex and Release through the Membrane. k_tot_	Dissociation of the Complex. k_diss_	
Levofloxacin free	21 ± 3	0.62 ± 0.06	10.4 ± 1.3	-	-
Lev in HPCD-PEI35-Man-**5**	32 ± 2	17 ± 1	1.5 ± 0.2	1.8 ± 0.2	40/60
Lev in HPCD-spermine-Man-**7**	43 ± 3	66 ± 5	2.1 ± 0.3	2.6 ± 0.3	60/40
Lev in HPCD-PEI1.8-Man-**8**	40 ± 2	30 ± 2	0.88 ± 0.07	0.96 ± 0.11	50/50
Lev in HPCD-PEI10-Man-**9**	44 ± 2	70 ± 2	1.15 ± 0.12	1.3 ± 0.2	65/35
Eugenol free	9 ± 1	0.36 ± 0.04	19.3 ± 2.4	-	-
EG in HPCD-PEI35-Man-**5**	16 ± 2	6.5 ± 0.5	4.6 ± 0.4	6.0 ± 0.5	30/70
EG in HPCD-spermine-Man-**7**	27 ± 3	4.5 ± 0.6	5.4 ± 0.4	7.5 ± 0.6	25/35
EG in HPCD-PEI1.8-Man-**8**	21 ± 3	25 ± 3	5.2 ± 0.3	7.1 ± 0.5	40/60
EG in HPCD-PEI10-Man-**9**	300 ± 20	57 ± 4	2.6 ± 0.2	3.0 ± 0.3	55/45

**Table 5 pharmaceuticals-15-00625-t005:** Diameters of zones of antibacterial action of complex formulations of levofloxacin and adjuvants (eugenol and apiol). *E. coli* (NCIB 12210). pH 7.4 (0.01 M PBS), 37 °C, 24 h of incubation. Molar ratios are given in parentheses (conjugate by HPCD’s tori). Values are presented as the mean± SD of three experiments.

Sample	Inhibition Zone Diameters (mm, ±0.5 mm) and Corresponding Concentration of Drug			
	0.5 μg/mL Lev	1 μg/mL Lev	2 μg/mL Lev	5 μg/mL Lev
Levofloxacin	13	22.5	28	31
HPCD-spermine-Man-**7** + Lev (1:1)	14	24	31	32
HPCD-spermine-Man-**7** + Lev/EG (1:5:5)	15	24.5	30.5	33
HPCD-PEI35-Man-**5** + Lev (1:1)	13	23	27	32
HPCD-PEI35-Man-**5** + Lev/EG (1:1:1)	14.5	25	27.5	34

## Data Availability

The data presented in this study are available in the main text and Appendix A.

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
