# Peer review of "Spectroscopy Approach for Highly-Efficient Screening of Lectin-Ligand Interactions in Application for Mannose Receptor and Molecular Containers for Antibacterial Drugs"

_pharmaceuticals, 2022, doi:10.3390/ph15050625_

Round 1
Reviewer 1 Report
Recommendation:Reject
In the manuscript, a class of molecular containers based on cyclodextrin, polyethyleneimine or spermine was prepared. Subsequently, a series of chemical structures, ligand affinity, and drug-carrying capacity were analyzed and verified for these molecular containers. After the model drug levofloxacin was encapsulated in different molecular containers, the drug loading efficiency and the release time of drug were increased. However, the methods for verifying the chemical structure of such molecular containers were relatively single, and the physicochemical evaluation of the final formulation was insufficient. In addition, the actual targeting of macrophages by molecular containers has not been demonstrated. Therefore, this manuscript is not sufficient for publication in this journal. Key questions about the design, experiments, and results are listed below:
- What are the advantages of the new analytical method used in the chemical identification section of this manuscript compared to existing analytical methods? It is suggested to supplement the comparison with other analytical methods.
- The title of the manuscript was “Molecular containers based on polyethyleneimine, spermine and cyclodextrins for targeted delivery of levofloxacin enhanced with adjuvant to alveolar macrophages”, but the targeting ability of formulations to macrophages was not verified in this Is this title appropriate? It is recommended to supplement the verification experiment of the macrophage targeting ability of such formulations.
- In the manuscript, it was not sufficient to rely on FTIR spectroscopy alone to analyze the chemical structure of the synthesized molecular containers. It is recommended to add other methods for confirmation, such as NMR, MS, etc.
- A series of molecular containers with different chemical structures were synthesized in the manuscript, resulting in different formulations when combined with drugs and adjuvants. Please indicate which formulation is beneficial for subsequent studies and what is the formulation of the optimal formulation.
- Quantitative data of drug encapsulation efficiency and loading efficiency should be clarified, and the analytical methods used in this part need to be validated, such as standard curve, detection limit, quantification limit and other pharmaceutical methodological parameters that need to be supplemented.
- It was mentioned that “levofloxacin and the promising adjuvant eugenol would reduce the dosage of the main "poisonous" drug”, so the optimal synergistic ratio of drugs and adjuvants should be investigated.
- It is mentioned that “cyclodextrins protect the drug from oxidation, hydrolysis, enzymatic damage, excessive moisture absorption”, but the stability of the final formulation in vitro and in vivo was not investigated. Please add on.
- Please elaborate on the reliability of using the homology model ConA instead of CD205.
- Related research on the effect of molecular containers on drug efficacy should be added to the manuscript.
It is recommended that the English in the manuscript need to be modified.
Author Response
Dear colleagues! The authors of the presented work sincerely thank you for thorough analysis of the manuscript and constructive comments! All of them have been addressed. Below you can see comments, answers to the reviewer's question and description of changes in the manuscript made by the authors during the revision.
With respect,
Zlotnikov I., Kudryashova E.
- What are the advantages of the new analytical method used in the chemical identification section of this manuscript compared to existing analytical methods? It is suggested to supplement the comparison with other analytical methods.
Surprisingly enough, there are very few groups in the world, who process the FTIR spectra quantitatively, to yield just as much molecular information as the method is designed to deliver. In fact, no other method allows to simultaneously track a variety of individual structural features (or types of features) with the simplicity of conventional UV/Vis spectrophotometer. NMR can resolve molecular features with greater detail, but it has very serious limitations regarding sample preparation and sample quantity, and it does not say much about intermolecular interactions. On the other end of the range, plasmon resonance is very sensitive to intermolecular interactions in real-time, but provides no insight as to which functional groups are involved in the interaction and whether or not they undergo any structural perturbations. FTIR, complemented by Fluorescence spectroscopy (anisotropy and quenching), just delivers the best of the two extremes. We did not invent any of those methods, but we suggest the efficient way of using them to solve some practical biochemical tasks.
The Table below summarizes and compares the key characteristics of the methods, commonly used to analyze the molecular interactions.
Criteria / method |
FTIR spectroscopy |
Fluorescence anisotropy + quenching |
Isothermal calorimetry |
NMR |
Surface plasmon resonance |
Information about functional groups, chemical bonds |
++ |
–/+ |
– |
+ |
–+ |
Molecular weight |
– |
+ |
– |
– |
–+ |
Secondary structure of protein-receptor |
++ |
– |
– |
+– |
– |
Mechanism of interaction of ligand and receptor |
+ |
– |
– |
+ |
– |
Energetic parameters |
+ |
+ |
++ |
– |
+ |
Phase transition |
+ |
– |
– |
– |
- |
Sensitivity |
–/+ |
+ |
– |
–/+ |
+ + |
- The title of the manuscript was “Molecular containers based on polyethyleneimine, spermine and cyclodextrins for targeted delivery of levofloxacin enhanced with adjuvant to alveolar macrophages”, but the targeting ability of formulations to macrophages was not verified in this Is this title appropriate? It is recommended to supplement the verification experiment of the macrophage targeting ability of such formulations.
Indeed, experiments with macrophages and their receptors have not been conducted – we are planning it for the future. The purpose of this research was more methodological than practical. We aimed at development of Spectroscopy Approach for High-Throughput Screening of Lectin-Ligand Interactions in Applications for the mannosilated conjugates Based on Cyclodextrins, with a model mannose receptor ConA as an example and proof-of-principle.
We agree that the title was a bit mis-leading, so we’ve changed it as follows: “Spectroscopy Approach for High-throughput Screening of Lectin-Ligand Interactions Using Concanavalin A as a Model Mannose Receptor in application for Molecular Containers for Antibacterial Drugs Based on Mannosylated Cyclodextrins”
- In the manuscript, it was not sufficient to rely on FTIR spectroscopy alone to analyze the chemical structure of the synthesized molecular containers. It is recommended to add other methods for confirmation, such as NMR, MS, etc.
We agree that a combination of methods is more appropriate (although we believe FTIR is very good for the purpose). In fact, we have used a couple of other methods as well, and we further amended the manuscript with NMR and HPLC data, as per the reviewer comments. Namely, we studied the structure of the conjugates by proton NMR spectroscopy (The results on NMR spectroscopy were added in the MS, section 2.1. HPCD-PEI-Man & HPCD-spermine-Man conjugates synthesis and characterization and Figure 2). The results obtained reconfirmed the proposed structure of the ligands studied. Complementary information to FTIR and NMR spectroscopy methods was obtained by spectrophotometry (with TNBS) and by fluorescence method (with OPA), allowing to determine the number of amino groups in the molecules at the synthesis stages (the results were added to the MS, Figure S25) which allowed us to judge the degree of modification/crosslinking. HPLC was carried out to purify the conjugates as well as to estimate their molecular weight (the results was added in the MS, Figure S24). The zeta potentials and hydrodynamic radii of the particles were determined by the method of dynamic light scattering. The combination of methods allows better judgement about the structure of conjugates. Mass spectra of the conjugates could not be registered successfully due to difficulties with ionization of samples.
- A series of molecular containers with different chemical structures were synthesized in the manuscript, resulting in different formulations when combined with drugs and adjuvants. Please indicate which formulation is beneficial for subsequent studies and what is the formulation of the optimal formulation.
We consider the most optimal is star-shaped or mesh conjugate of CD with spermine or light branched PEI (conjugates 7-9 in table 2). For example, the high-affinity conjugate CD-spermine-7: has one of the highest Affinities to mannose receptors, a long release period of the drug (Lev and EG), and high Entrapment efficiency ЕЕ as well as acceptable loading capacity (LC).
Table 2
Sample's designation * |
Prolonged action of drug |
Entrapment efficiency |
Affinity to mannose receptors |
HPCD-spermine-Man-1 |
++ |
++ |
+ |
HPCD-spermine-Man-2 |
++ |
++ |
+ |
HPCD-spermine-Man-3 |
++ |
+ |
+++ |
HPCD-PEI35-Man-4 |
+ |
+ |
+ |
HPCD-PEI35-Man-5 |
++ |
++ |
+ |
HPCD-PEI35-Man-6 |
++ |
++ |
++ |
HPCD-spermine-Man-7 |
+++ |
++ |
+++ |
HPCD-PEI1.8-Man-8 |
++ |
++ |
++ |
HPCD-PEI10-Man-9 |
+++ |
++ |
++ |
- Quantitative data of drug encapsulation efficiency and loading efficiency should be clarified, and the analytical methods used in this part need to be validated, such as standard curve, detection limit, quantification limit and other pharmaceutical methodological parameters that need to be supplemented.
The required data was added to the manuscript (section 2.3.2.1. and section 3.10.): the Loading capacity was calculated as well as the concentration of levofloxacin conjugated to container -separately and in combination with eugenol (EG). The information is presented in Table 3 and discussed in lines 503-532 in the MS.
Briefly: The Loading capacity and entrapment efficiency were calculated based on the values of dissociation constants and the number of Lev or EG molecules loaded by one container molecule, which in turn were calculated based on spectral data (FTIR): by shifting the position of the maximum of the 2nd derivative of the aromatic peak, as well as by the intensity of the peak itself. Fitting of the curves of change of the analytical signal ξ versus molar excess of the ligand x was carried out using the Hill and Boltzmann equations; linearization in Hill’s coordinates was completed (as described in MS in section 3.10). EE for Lev was 85-98%, for eugenol 60-80%. Loading capacity for Lev is also quite high (mostly in the range of 53-86% (w/w)).
- It was mentioned that “levofloxacin and the promising adjuvant eugenol would reduce the dosage of the main "poisonous" drug”, so the optimal synergistic ratio of drugs and adjuvants should be investigated.
A dedicated article was written by the authors (“Essential oils as antibacterial agents and levofloxacin synergists” by ID. Zlotnikov, NG. Belogurova, SS. Krylov, MN. Semenova, VV. Semenov and EV. Kudryashova [1]) which is devoted to the synergy effect, currently under review by the journal Pharmaceuticals. Using the agar diffusion method, we shown that the minimum inhibitory concentration (MIC) of levofloxacin on E.coli is 0.1-0.15 μg/ml, reduced to 0.06 μg/ml, for the levofloxacin-cyclodextrin complex presumably due to increased penetration through the cell membrane. Essential oils in soluble form have weak antibacterial activity at concentrations of 2-5 mg/ml, however, apiol and eugenol can significantly enhance the inhibitory effect of levofloxacin presumably due to inhibition of efflux by interaction with P-glycoprotein [1]. The most striking synergism was demonstrated by eugenol, significantly reducing the MIC (against E.coli) of levofloxacin from 0.1-0.06 to 0.02-0.03 μg/ml. The inhibitory effect of levofloxacin in free form on B.subtilis begins at a concentration of 0.45 μg/ml, and the addition of 2-3.5 mg/ml of the adjuvant reduces the MIC to 0.15 μg/ml Lev. This opens the prospects for the creation of combined enhanced fluoroquinolone drugs to enhance the therapeutic effect and reduce the toxic effect of the drug on the body. Finding optimal ratio for practical applications would still need a significant amount of work, with involvement of clinically relevant pathogens (not just model bacteria).
- It is mentioned that “cyclodextrins protect the drug from oxidation, hydrolysis, enzymatic damage, excessive moisture absorption”, but the stability of the final formulation in vitro and in vivo was not investigated. Please add on.
The formation of Lev complexes with conjugates studied results in Lev protection from external influence (such as ultraviolet light and biodegradation). The corresponding section has been added to the manuscript: section 2.4. Antibacterial activity of levofloxacin and adjuvants in molecular containers.
The stability of formulations studied of levofloxacin and eugenol in conjugates in vitro by monitoring the level of antibacterial activity of the formulations: the prolonged effect of 6 days has been shown for formulations studied in experiments on bacterial cells, which confirms the stability of the drug formulation in the composition of the molecular container, as well as its effectiveness. The decrease in Lev antibacterial activity after the 4th day might be explained by the biodegradation of drug molecule. For instance, the authors [A.S. Maia, A.R. Ribeiro, C.L. Amorim, M.E. Tiritan, Degradation of fluoroquinolone antibiotics and identification of metabolites/transformation products by liquid chromatography-tandem mass spectrometry, J. Chromatogr. A. 1333 (2014) 87–98.] demonstrate the chemical changes in the structure of fluoroquinolone's pharmacophore (the cleavage of the fluorine atom as well as the oxidation and/or damage of the heterocycle) by bacterial strains.
- Please elaborate on the reliability of using the homology model ConA instead of CD205.
The relevance of ConA as a CD206 model has been widely discussed in the literature and studied by us through computer modeling of 15 ligands (https://doi.org/10.1134/S0006297922010059). ConA is an easily accessible lectin having a carbohydrate-binding domain structure similar to CD206 and a general similarity in carbohydrate-binding ability [8,17–21 in MS]. According to the literature data, the CD206 domain shows a high similarity with mannan binding lectin A, which, in turn, is similar to ConA in terms of carbohydrate binding [22 in MS]. This is confirmed by numerous in vivo and in vitro studies [20]:
Mannose-containing ligands, pre-selected by ConA assay, show high affinity (by flow cytometry) to CD206+ cells [23,24], similarly, mannose-containing ligands, pre-selected by ConA assay, are absorbed by macrophages via receptor-mediated internalization mechanism [7], and bio-distribute to lungs due to adsorption by macrophages [25]. All this data support the relevance of using ConA as macrophage receptor model. We have previously shown [17] that ConA is a relevant CD206 (CRD4) model due to the similarity of the structural organization of the binding sites and the high correlation of the values of the free energies of complexation r > 0.9. The similarity in the processes of complexation of ConA and CD206 with carbohydrate ligands was revealed by molecular dynamics calculation: in both cases the interaction is stabilized by the electrostatic interactions of charged residues (Asn, Asp, Arg) with oxygen and hydrogen atoms in carbohydrates and additionally by CH-π stacking interactions of Tyr/Trp with the planes of Man residues.
Thus, in this study we have demonstrated the possibility to deftly performing primary screening of a wide range of ligands (potentially to any receptor) without using cells or animals.
- Related research on the effect of molecular containers on drug efficacy should be added to the manuscript.
Experimental data on antibacterial activity of the developed multifunctional systems on gram(-) bacterial cells was added to the manuscript, demonstrating the potential of molecular containers for creating complex enhanced formulations of antibacterial drugs in order to reduce the dosage of the toxic agent, as well as to prolong its action. Bacterial growth was suppressed by all samples with Lev for 24 hours. EG (eugenol) and apiol enhance and accelerate the antibacterial effect of Lev in conjugates. Lev in the composition of conjugates acts 50% more effectively than Lev in the free form. The corresponding section has been added to the manuscript: 2.4. Antibacterial activity of levofloxacin and adjuvants in molecular containers.
The authors are grateful to the reviewer for valuable comments.

Reviewer 2 Report
Comments: Manuscript is very promising and smart work. Few issues should to be studied to clarify such this interesting work.
- The main aim of this manuscript is to create molecular container for being used to target Alveolar Macrophages. However, cultured Alveolar Macrophages were not used. Targeting capacity was not studied. Cytotoxicity was not evaluated.
- FITC labelled ligands used in the manuscript and was not measured by confocal microscopy to confirm accumulation of container in perinuclear region. Authors must to do this experiment.
- Loading capacity was not calculated. The concentration of levofloxacin conjugated to container separated or in combination with eugenol should to be calculated.
- The release studies must be performed in PBS pH 7.4 and 6.5 . additionally, the cumulative drug release % should to be measured for more than 48h incubation.
- Authors did not provide description for how such this molecular container can adhere cell membrane more easier than mucoadhesive polymers such as chitosan
Author Response
Dear colleagues! The authors of the presented work sincerely thank you for thorough analysis of the manuscript and constructive comments! All of them have been addressed. Below you can see comments, answers to the reviewer's question and description of changes in the manuscript made by the authors during the revision.
With respect,
Zlotnikov I., Kudryashova E.
- The main aim of this manuscript is to create molecular container for being used to target Alveolar Macrophages. However, cultured Alveolar Macrophages were not used. Targeting capacity was not studied. Cytotoxicity was not evaluated.
- Indeed, experiments with macrophages and their receptors have not been conducted – we are planning it for the future.
The purpose of this research was more methodological than practical. We aimed at development of Spectroscopy Approach for High-Throughput Screening of Lectin-Ligand Interactions in Applications for the mannosilated conjugates Based on Cyclodextrins, with a model mannose receptor ConA as an example and proof-of-principle.
It should be mentioned that due to the limited availability of the CD206 receptor [13], the range of applicable methods is limited, also high throughput screening analysis of ligands is not available. Accordingly, in such cases, a relevant model protein ConA can be used, which is more suitable for primary screening purposes. The relevance of ConA as a CD206 model has been widely discussed in the literature and studied by us through computer modeling of 15 ligands (https://doi.org/10.1134/S0006297922010059). ConA is an easily accessible lectin having a carbohydrate-binding domain structure similar to CD206 and a general similarity in carbohydrate-binding ability [8,17–21]. According to the literature data, the CD206 domain shows a high similarity with mannan binding lectin A, which, in turn, is similar to ConA in terms of carbohydrate binding [22]. This is confirmed by numerous in vivo and in vitro studies [20]:
Mannose-containing ligands, pre-selected by ConA assay, show high affinity (by flow cytometry) to CD206+ cells [23,24], similarly, mannose-containing ligands, pre-selected by ConA assay, are absorbed by macrophages via receptor-mediated internalization mechanism [7], and bio-distribute to lungs due to adsorption by macrophages [25]. All this data support the relevance of using ConA as macrophage receptor model. We have previously shown [17] that ConA is a relevant CD206 (CRD4) model due to the similarity of the structural organization of the binding sites and the high correlation of the values of the free energies of complexation r > 0.9. The similarity in the processes of complexation of ConA and CD206 with carbohydrate ligands was revealed by molecular dynamics calculation: in both cases the interaction is stabilized by the electrostatic interactions of charged residues (Asn, Asp, Arg) with oxygen and hydrogen atoms in carbohydrates and additionally by CH-π stacking interactions of Tyr/Trp with the planes of Man residues.
Thus, we have demonstrated the possibility to deftly performing primary screening of a wide range of ligands (potentially to any receptor) without using cells or animals.
The title of the article has been changed: “Spectroscopy approach for High-throughput screening of lectin-ligand interactions using concanavalin A as a Model mannose receptor in application for Molecular Containers for antibacterial drugs Based on Cyclodextrins, Oligoamines and Mannose”
Cytotoxicity was not evaluated
Experimental data on antibacterial activity of the developed multifunctional systems on gram(-) bacterial cells was added to the manuscript, demonstrating the potential of molecular containers for creating complex enhanced formulations of antibacterial drugs in order to reduce the dosage of the toxic agent, as well as to prolong its action. The corresponding section has been added to the manuscript: 2.4. Antibacterial activity of levofloxacin and adjuvants in molecular containers.
- FITC labelled ligands used in the manuscript and was not measured by confocal microscopy to confirm accumulation of container in perinuclear region. Authors must to do this experiment.
Indeed, FITC was used in this work, however, that was an experiment to measure the fluorescence anisotropy of the labeled ligand upon ligand-receptor interaction. So, it was purely “chemical” assay, with no living cells involved.
At the same time, here we made cell survival experiments, showing that levofloxacin (lev) conjugated with mannose-rich derivatives of cyclodextrins stops bacterial growth about 50% more effectively than conventional lev. Which actually means it can make it to the nucleus, as the lev’s molecular target, bacterial DNA gyrase, is located in the nucleus. We are planning more advanced cell experiments in our future work – and confocal microscopy would likely be a method of choice to look at the intracellular localization.
Increased antibacterial activity of the conjugates can be explained by enhanced penetration through the cell membrane [40] and by increased adhesion of conjugates to the cell wall, as we have recently shown using electronic microscopy [37].
- Loading capacity was not calculated. The concentration of levofloxacin conjugated to container separated or in combination with eugenol should to be calculated.
The required data was added in the manuscript (section 2.3.2.1.): the Loading capacity was calculated as well as the concentration of levofloxacin conjugated to container separated and in combination with eugenol. The information is presented in Table 3 and discussed in lines 503-532 in the MS.
Briefly: The Loading capacity and entrapment efficiency were calculated based on the values of dissociation constants and the number of Lev or EG molecules loaded by one container molecule, which in turn were calculated based on spectral data (FTIR): by shifting the position of the maximum of the 2nd derivative of the aromatic peak, as well as by the intensity of the peak itself. EE for Lev was 85-98%, for eugenol 60-80% (at with a threefold excess of cyclodextrin tori in relation to LF). Loading capacity is also quite high and it decreases with increasing molecular weight of the container, but still exceeds 25% (w/w).
- The release studies must be performed in PBS pH 7.4 and 6.5 additionally, the cumulative drug release % should to be measured for more than 48h incubation.
We have amended the manuscript with this data. The release studies (using dialysis with 12kDa MWCO membrane) for our formulations of levofloxacin and eugenol in the complexes with conjugates was performed in PBS pH 7.4 and 6.5, with increased duration of experiments. The effective time to achieve equilibrium of Lev and EG release was in the range of 25-70 hours for most carriers, as compared to 30 min for non-conjugated Lev. There were no noticeable differences in the release rates at pH 6.5 and 7.4 (less than 10% - within the experimental error). However, a significant increase in the dissociation rate is achieved in an acidic medium at pH 2-3 (t1/2 does not change, and t80% decreases to 5-20 hours).
- Authors did not provide description for how such this molecular container can adhere cell membrane more easier than mucoadhesive polymers such as chitosan
In fact, we do not think that our molecular containers are more mucoadhesive (or cell membrane adhesive) than chitosan. Based on the literature, we believe they have comparable mucoadhesive parameters [1 and 5]. Although, head-to-head comparison would have been more useful to judge this. The reason why they should be comparable – is that they have similar chemical structure (carbohydrate polymers), and a similar charge density (dzeta potential), which are the key factors for mucoadhesiveness.
References
- Sogias, I.A., Williams, A.C., and Khutoryanskiy, V.V., Biomacromolecules, 2008, vol. 9, pp. 1837–1842.
- Bowman, R. and Leong, K.W., Int. J. Nanomedicine, 2006, vol. 1, pp. 117–128.
- Garg, T., Rath, G., and Goyal, A.K., Artif. Cells Nanomed. Biotechnol., 2015, pp. 1–5.
- Asthana G.S., Asthana A., Kohli D.V., Vyas S.P. // BioMed Res. Int. 2014. V. 2014. P. 1–17.
- Gou M.L., Dai M., Li X., Yang L., Huang M., Wang Y.,Kan B., Lu Y., Wei Y., Qian Z. // Coll. Surf. B Biointer. 2008. V. 64. P. 135–139.
The authors are grateful to the reviewer for valuable comments.
Reviewer 3 Report
The manuscript proposes an interesting research work focused on the formulation of polyethyleneimine, spermine and cyclodextrins for drug delivery purposes.
The topic is appropriate for the journal.
The title is adequate and correlate with the content of the article.
The abstract reports a consistent summary of the article findings.
The work has a clear structure.
All sections are properly written and required for a complete understanding.
Nevertheless, there are minor issues that require to be addressed before proceeding with the publication, to enhance the quality and presentation to a broad audience.
References are appropriately mentioned, but it worths mentioning that the paper would benefit a lower self-citation rate by the authors.
Additional references over introduction section would boost the manuscript by mentioning a more complete overview of a broader plethora of attempted approaches/materials in order to emphasize the scientific soundness of the presented findings (Szejtli, J., 1998, Chem. Rev. 98, 1743–1754; Milcovich et. al., Int. J. Pharm, 2018, 548, 474–479; Connors, K.A., 1997, Chem. Rev. 97,1325–1357.).
The conclusion section would definitely benefit further explanation, e.g. addition of a few sentences recapitulating the whole findings, the scientific progress and soundness of the original research work.
It is suggested to add on the Supplementary Materials file a guiding index , with a list of figures, followed by the figures already present on the file.
Author Response
Dear colleagues! The authors of the presented work sincerely thank you for thorough analysis of the manuscript and constructive comments! All of them have been addressed. Below you can see comments, answers to the reviewer's question and description of changes in the manuscript made by the authors during the revision.
With respect,
Zlotnikov I., Kudryashova E.
References are appropriately mentioned, but it’s worth mentioning that the paper would benefit a lower self-citation rate by the authors.
Links to the authors' articles are provided to explain and confirm many of the parameters discussed:
To clarify quantitative data on the effectiveness of encapsulation of the drug and the effectiveness of conjugate loading, methods of determination are given. Analytical methods are described.
Additional references over introduction section would boost the manuscript by mentioning a more complete overview of a broader plethora of attempted approaches/materials in order to emphasize the scientific soundness of the presented findings (Szejtli, J., 1998, Chem. Rev. 98, 1743–1754; Milcovich et. al., Int. J. Pharm, 2018, 548, 474–479; Connors, K.A., 1997, Chem. Rev. 97,1325–1357.).
Recommended references have been included in the list of references, to emphasize the scientific soundness of the presented findings. Thank you for the valuable material!
The conclusion section would definitely benefit further explanation, e.g. addition of a few sentences recapitulating the whole findings, the scientific progress and soundness of the original research work.
Recapitulation of the findings, the scientific progress and soundness of the original research work was added into conclusion section
It is suggested to add on the Supplementary Materials file a guiding index , with a list of figures, followed by the figures already present on the file.
A guiding index with a list of figures was added In the Supplementary Materials file.
The authors are grateful to the reviewer for valuable comments.
Round 2
Reviewer 1 Report
Recommendation: Minor revision
The authors have made great efforts to address our comments in the revised manuscript. However, some questions still existed and were not fully answered. Detailed suggestions are listed below:
- For question 1: The title of “Spectroscopy Approach for High-throughput Screening of Lectin-Ligand Interactions Using Concanavalin A as a Model Mannose Receptor in application for Molecular Containers for Antibacterial Drugs Based Cyclodextrins” was too long and complicated, please simplify.
- For question 4: The answer mentioned that “the high-affinity conjugate CD-spermine-7: has one of the highest affinities to mannose receptors, a long release period of the drug (Lev and EG), and high entrapment efficiency (ЕЕ) as well as acceptable loading capacity (LC)”. The detailed description of the molecular container formed by this conjugate need to be added in the manuscript.
- For question 5: It is strongly recommended that the reliability validation data on drug content determination should to be supplemented, such as precision, accuracy, recovery, etc.
- For question 6: Even though other articles are pending review, we recommend that this manuscript detail the exact ratio of the two drugs used and the reason for that ratio.
Author Response
Dear Reviewer! The authors of this paper sincerely thank you for attentive and constructive review! Your comments made the manuscript better. Please find our responses to the latest series of questions.
Best wishes,
Zlotnikov I., Kudryashova E.
- For question 1: The title of “Spectroscopy Approach for High-throughput Screening of Lectin-Ligand Interactions Using Concanavalin A as a Model Mannose Receptor in application for Molecular Containers for Antibacterial Drugs Based Cyclodextrins” was too long and complicated, please simplify.
The title has been simplify: “Spectroscopy Approach for Highly-Efficient Screening of Lectin-Ligand Interactions in Application for Mannose Receptor and Molecular Containers for Antibacterial Drugs”.
- For question 4: The answer mentioned that “the high-affinity conjugate CD-spermine-7: has one of the highest affinities to mannose receptors, a long release period of the drug (Lev and EG), and high entrapment efficiency (ЕЕ) as well as acceptable loading capacity (LC)”. The detailed description of the molecular container formed by this conjugate need to be added in the manuscript.
The high-affinity conjugate CD-spermine-7: The manuscript provides a detailed description of the leading conjugates 7 (star-shaped cyclodextrin with grafted spermines): the synthesis of HPCD-spermine-Man conjugate and its characterization including physical-chemical properties is presented (section 2.1., Figure 1. (a), Table 1.). The successful formation of the conjugate CD-spermine-7 was demonstrated by Fourier transform infrared (FTIR) spectroscopy, as shown in Fig.2. The structure of the synthesized conjugates was confirmed by NMR spectroscopy (Fig. 2d). The degree of mannosylation was determined by two methods - by titration of amino groups in conjugates with TNBS, and by fluorimetric titration of amino groups with OPA (Fig. S25). Dissociation constants of conjugate with ConA–ligand complexes were determined. Release kinetics curves of levofloxacin and eugenol in the complexes with molecular container 7-9 are presented (in Fig. 7) in comparison with free drugs. Detailed comments on the entrapment efficiency (ЕЕ) and loading capacity of this conjugates for levofloxacin and eugenol have been added to the section 2.3.2. (Table 3). Conjugate 7 demonstrated the highest parameters of loading capacity and entrapment efficiency to levofloxacin and eugenol. This conjugate combines the valuable properties of a drug container and a high affinity to mannose receptors due to presence of the trimannoside analogue on spermine spacer. The explanation was added to the section 2.3.2.1 (line 506-510), and to Conclusion.
- For question 5: It is strongly recommended that the reliability validation data on drug content determination should to be supplemented, such as precision, accuracy, recovery, etc.
The drug content in the conjugates, as well as entrapment efficiency (ЕЕ), loading capacity (LC), and dissociation constants of ConA–ligand complexes (Kd), are presented (Tables 1-3) with confidence intervals calculated using the Student's test (P=0.95, n=3). These parameters were determined from the dependencies of analytical signals (peak intensity in the IR spectra, peak position in the spectrum, fluorescence intensity) on the titrant concentration which was approximated by the Hill equation (coefficients R2, were at least 0.98, and often more than 0.99). The resulting parameters (entrapment efficiency (ЕЕ) and loading capacity (LC), dissociation constants of ConA–ligand complexes (Kd)), have maximal error values of 10%. The corresponding precision and accuracy data was added to the manuscript.
- For question 6: Even though other articles are pending review, we recommend that this manuscript detail the exact ratio of the two drugs used and the reason for that ratio.
In this article, we just took 1:1 and 1:2 ratios, without any optimization. We aimed at showing the proof-of-principle – that there is a possibility as such to obtain the double complexes of Levofloxacin and its adjuvants (essential oils), and to study their parameters, ligand binding constants and release kinetics of drug molecules from both single and double complexes. We have demonstrated that eugenol and apiol exhibit a synergy effect with Levofloxacin, boosting its antibacterial activity at concentrations several orders of magnitude lower than is needed for these substances to show any antibacterial activity [2].
In another paper we submitted for publication in Pharmaceuticals, the ratio of Levofloxacin:eugenol = 1:500 – 1:1000 was found to be optimal, yielding antibacterial activity of Levofloxacin at concentrations several fold lower than it can be normally expected. Other authors [1] have reported similar synergistic effect of geraniol, carveol and other terpenoids, with a similar antibiotic:enhancer ratio (1:1000).
Relevant comments have been added to the manuscript.
References
- Pereira de Lira, M.H.; Fernandes Queiroga Moraes, G.; Macena Santos, G.; Patrício de Andrade Júnior, F.; De Oliveira Pereira, F.; Oliveira Lima, I. Synergistic Antibacterial Activity of Monoterpenes in Combination with Conventional Antimicrobials against Gram-Positive and Gram-Negative Bacteria. Rev. Ciências Médicas e Biológicas 2020, 19, 258, doi:10.9771/cmbio.v19i2.33665.
Reviewer 2 Report
Authors have addressed comment of reviewer point by point and manuscript is more acceptable NOW.
Author Response
Dear Reviewer, thank you for your thoughtful review and constructive comments, they were very helpful to make the manuscript better.
Sincerely.